# Multi-level processing of emotions in life motion signals revealed through pupil responses

Tian Yuan[1,2], Li Wang[1,2]*, Yi Jiang[1,2]

[1]State Key Laboratory of Brain and Cognitive Science, Institute of Psychology, Chinese Academy of Sciences, Beijing, China; [2]Department of Psychology, University of Chinese Academy of Sciences, Beijing, China

**Abstract** Perceiving emotions from the movements of other biological entities is critical for human survival and interpersonal interactions. Here, we report that emotional information conveyed by point-light biological motion (BM) triggered automatic physiological responses as reflected in pupil size. Specifically, happy BM evoked larger pupil size than neutral and sad BM, while sad BM induced a smaller pupil response than neutral BM. Moreover, this happy over sad pupil dilation effect is negatively correlated with individual autistic traits. Notably, emotional BM with only local motion features retained could also exert modulations on pupils. Compared with intact BM, both happy and sad local BM evoked stronger pupil responses than neutral local BM starting from an earlier time point, with no difference between the happy and sad conditions. These results revealed a fine-grained pupil-related emotional modulation induced by intact BM and a coarse but rapid modulation by local BM, demonstrating multi-level processing of emotions in life motion signals. Taken together, our findings shed new light on BM emotion processing, and highlight the potential of utilizing the emotion-modulated pupil response to facilitate the diagnosis of social cognitive disorders.

*For correspondence:
wangli@psych.ac.cn

Competing interest: The authors declare that no competing interests exist.

## eLife Assessment

This **important** study provides **convincing** evidence that emotional information in biological motion can induce different patterns of pupil responses, which could serve as a behavioral marker of an autistic trait. These results broaden our understanding of how emotional biological motion can automatically trigger physiological changes and reveal the potential of using emotional-modulated pupil response to facilitate the diagnosis of social cognitive disorders. The work will be of broad interest to cognitive neuroscience, psychology, affective neuroscience, and vision science.

## Introduction

Perceiving and interpreting emotions from various social signals is critical for human social functioning, which enables us to infer the intentions of our conspecifics and further facilitates interpersonal interactions. Facial expressions present the most common non-verbal social communicative signals regarding others' affective states and intentions (*Frith, 2009*). In addition to faces, the movement of biological organisms serves as another essential type of social signal carrying significant emotional information, and this information remained salient even from a far distance (*de Gelder, 2006*). The human visual system is highly sensitive to such signals that we can readily decipher emotions from biological motion (BM), even when it was portrayed by several point lights attached to the major joints (*Johansson, 1973*; *Troje, 2008*). Moreover, it has been found that happy point-light walkers were recognized faster

and more accurately than sad, angry, or neutral walkers, demonstrating a happiness superiority (*Lee and Kim, 2017*; *Spencer et al., 2016*). While these studies provided some insights into the emotion processing mechanism of BM, they relied mainly on the explicit identification and active evaluation of BM emotions. Importantly, the encoding of emotional information also involves a rather automatic and implicit process that is independent of the participant's explicit identifications (*Critchley, 2000*; *Lange et al., 2003*; *Okon-Singer et al., 2013*; *Shafer et al., 2012*). Notably, this implicit aspect of emotion processing could be even more effective in probing individual differences and social deficits (*Kana et al., 2016*; *Keifer et al., 2020*; *Kovarski et al., 2019*; *Wong et al., 2008*), as it requires the intuitive processing of emotions that could not be learned (*Frith, 2004*). For example, research has shown that individuals with autistic disorders showed altered neural activities during implicit but not explicit emotion processing of natural scenes (*Kana et al., 2016*). These observations highlighted the importance of using objective measurements to investigate the implicit and automatic aspect of BM emotion processing.

The pupil response hence serves as a promising approach for reliably unfolding the implicit emotion processing of BM as it adds no extra task requirements to the cognitive process (*Burley et al., 2017*). In particular, pupil size is related to the activity of the automatic nervous system mediated by the locus coeruleus norepinephrine, which not only responds to physical light, but also reflects the underlying cognitive state (*Joshi and Gold, 2020*). Moreover, it could spontaneously capture the current subjective state and is thus useful for revealing the time course of the related cognitive processing (*de Gee et al., 2014*; *Graves et al., 2021*; *Kloosterman et al., 2015*; *Oliva and Anikin, 2018*). Recently, this measurement has been introduced to the field of emotion perception, and emerging evidence has suggested that emotions conveyed by social signals (e.g., faces, voices) could modulate pupil responses. For instance, happy and angry dynamic faces elicited a pupil dilation effect as compared with sad and neutral faces (*Burley and Daughters, 2020*; *Prunty et al., 2022*). Noticeably, the point-light BM, as another critical type of social signal, also conveyed salient affective information. Besides, its emotion processing mechanism is closely connected with that of faces (*Alaerts et al., 2011*; *Becker et al., 2011*; *Lee and Kim, 2017*; *Yuan et al., 2024*). However, it remains unequivocal whether the pupil also responds to the emotional information carried by the minimalistic point-light walker display. Such physiological observation can faithfully uncover the implicit emotion processing of BM, and it also expands the existing line of inquiry on the emotion processing of social cues.

To fill this gap, the current study implemented the pupil recording technique together with the passive viewing paradigm to explore the automatic and implicit emotion processing of BM. We first investigated the pupil responses to point-light walkers with different emotions (i.e., happy, sad, and neutral). In addition to intact emotional BM sequences, we also tested scrambled emotional BM sequences, which lack the gestalt of a global figure but preserve the same local motion components as the intact walkers. Recent studies have shown that such local motion signals play an essential role in conveying biologically salient information such as animacy, and walking direction (*Chang and Troje, 2008*; *Chang and Troje, 2009*; *Troje and Westhoff, 2006*). This study went further to investigate whether this local BM signal could convey emotional information and elicit corresponding pupil responses. Moreover, given that emotion perception from social signals is generally impaired in individuals with autism (*Harms et al., 2010*; *Hubert et al., 2007*), we also took the individual autistic traits into consideration by measuring the autistic tendencies in normal populations with the autistic quotient (AQ) questionnaire (*Baron-Cohen et al., 2001*).

## Results

### Experiments 1a and 1b: intact emotional BM

In Experiment 1a, we investigated whether emotional BM could automatically exert influences on pupil responses. Specifically, participants were instructed to passively view the intact happy/sad/neutral BM with their pupil responses being recorded simultaneously (see *Figure 1*). A cluster-based permutation analysis was applied to illustrate the time course of emotional modulations on pupil responses. Results showed that the happy BM induced a significant pupil dilation effect than the neutral BM from 1750 to 3200 ms (see *Figure 2A*). Conversely, the sad BM evoked a significantly smaller pupil response than the neutral BM, which starts from 1850 ms until the end of the stimulus presentation (see *Figure 2A*). Moreover, the happy BM evoked a significantly larger pupil response as compared

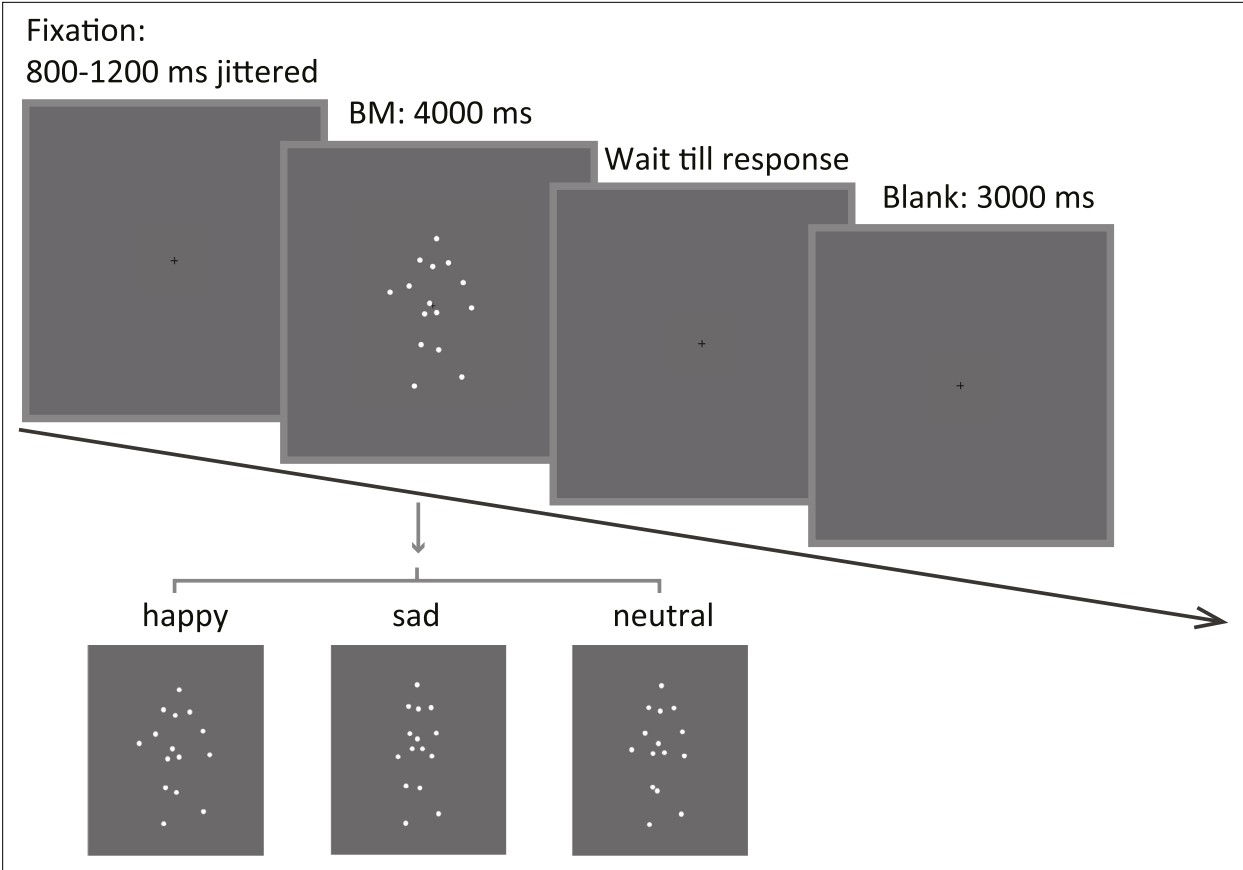

**Figure 1.** Schematic representation of the experimental procedure and the stimuli in Experiment 1a. A happy/sad/neutral biological motion (BM) walker turning 45° leftwards or rightwards was presented at the center of the screen for 4000 ms. Participants were instructed to maintain their fixation on the BM stimuli during stimulus presentation and to continue the procedure through key pressing.

to the sad BM in a longstanding time window, ranging from 1200 ms until the disappearance of the display (see *Figure 2A*).

We then computed the mean pupil size by collapsing the pupillometry across all time points, ranging from the onset of the stimuli to the end of the stimuli presentation. A one-way repeated measures analysis of variance (ANOVA) was conducted on the mean pupil size for each emotional condition (happy, sad, neutral), and the results showed a significant main effect of emotional condition ($F$(1.4, 32.2) = 8.92, p = 0.002, $\eta_p^2$ = 0.28; Greenhouse–Geisser corrected, *Figure 3A*). On average, the happy BM induced a significantly larger pupil response than the neutral BM ($t$(23) = 2.73, p = 0.024, Cohen's $d$ = 0.56, 95% confidence interval (CI) for the mean difference = [0.04, 0.27]; Holm-corrected, p = 0.036 after Bonferroni correction, *Figure 3A*). In contrast, the sad BM evoked a significantly smaller pupil size than the neutral BM (sad vs. neutral: $t$(23) = −2.43, p = 0.024, Cohen's $d$ = 0.50, 95% CI for the mean difference = [−0.35, −0.03]; Holm-corrected, p = 0.071 after Bonferroni correction, *Figure 3A*). Moreover, the happy BM evoked a significantly larger pupil size than the sad BM ($t$(23) = 3.34, p = 0.009, Cohen's $d$ = 0.68, 95% CI for the mean difference = [0.13, 0.55]; Holm-corrected, p = 0.009 after Bonferroni correction, *Figure 3A*), which echoes with former studies showing a happiness advantage in BM processing (*Actis-Grosso et al., 2015*; *Lee and Kim, 2017*; *Spencer et al., 2016*; *Yuan et al., 2023*). Importantly, the observed happy over sad dilation effect is negatively correlated with the individual autistic traits ($r$(22) = −0.47, p = 0.022, 95% CI for the mean difference = [−0.73, −0.08], *Figure 4A*), indicating a compromised ability to perceive emotions from BM among individuals with higher autism tendencies. No significant correlations were found between AQ and other pupil modulation effects (*Figure 4B, C*). Additionally, no significant correlations were observed between AQ and the original pupil responses induced by happy/neutral/sad BM (see *Figure 5A*). This is potentially because the original pupil response is a mixed result of stimuli perception and emotion perception,

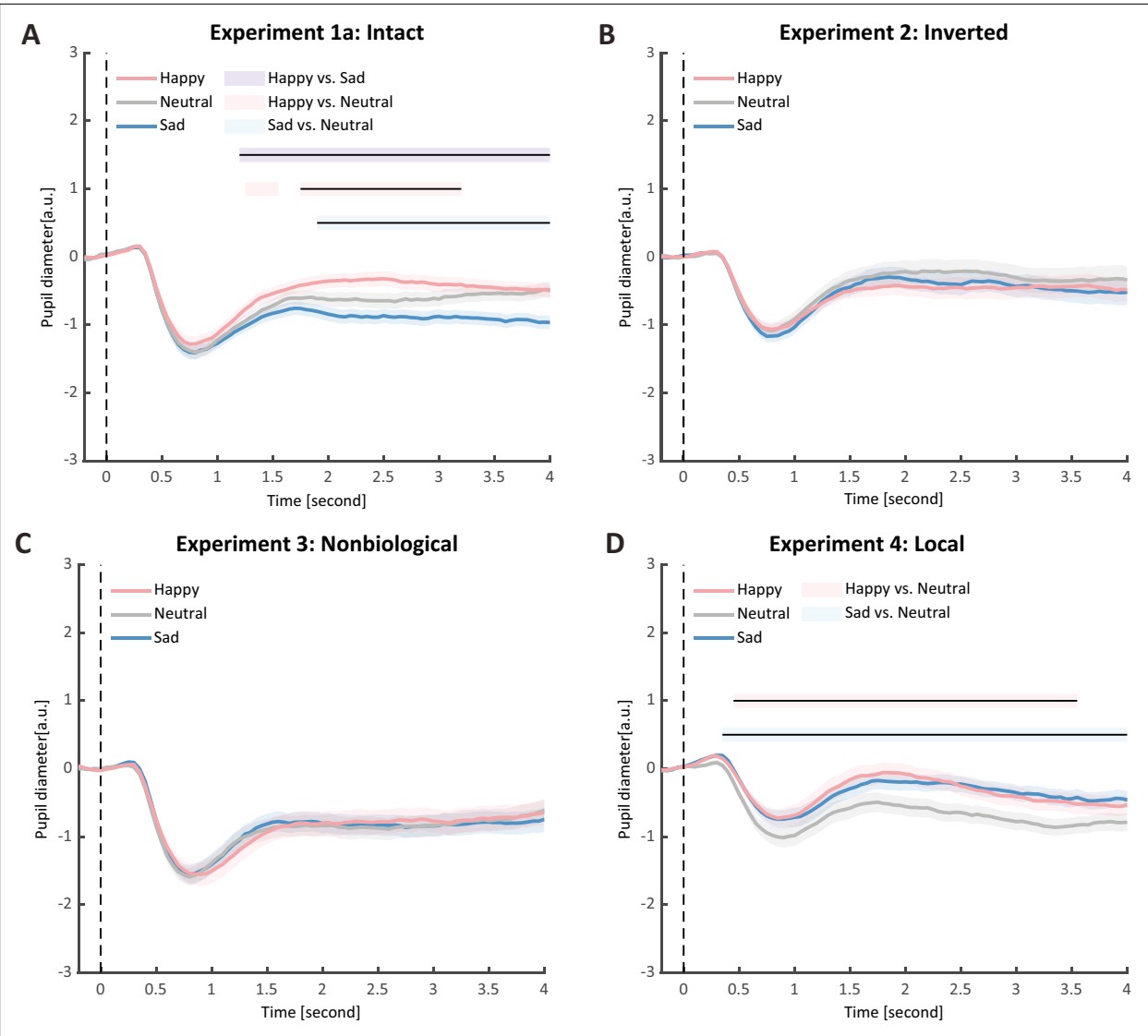

**Figure 2.** Time course of pupil responses to happy, sad, and neutral biological motion (BM) in Experiments 1–4. Solid lines represent pupil diameter under each emotional condition as a function of time (happy: red; sad: blue; neutral: gray); shaded areas represent the standard error of the mean (SEM) between participants (N = 24); colored horizontal lines indicate periods during which there are statistically significant differences among conditions at p < 0.05; and black horizontal lines indicate significant differences after cluster-based permutation correction. All the pupil data are in arbitrary units (a.u.) (**A**) In Experiment 1a, the happy BM evoked larger pupil response as compared to the sad and neutral BM, and the sad BM evoked smaller pupil size than the neutral BM. (**B**) In Experiment 2, the inverted BM failed to produce such emotional modulation effects. (**C**) In Experiment 3, the emotional BM that is deprived of the local motion feature exerted no emotional modulation on pupil responses. (**D**) In Experiment 4, both the happy and sad local BM induced a larger pupil response than the neutral local BM, and such dilation effect started from a relatively early time point.

while the pupil changes across emotional conditions could more faithfully reflect individual sensitivities to emotions in BM (*Burley et al., 2017*; *Pomè et al., 2020*; *Turi et al., 2018*).

To strengthen the correlation results, we conducted a replication experiment (Experiment 1b) and added a test–retest examination to further assess the reliability of our measurements. Specifically, a new group of participants were recruited to perform the identical task, and they were asked to return to the lab for a retest. The results again revealed the main effect of emotional condition in both the first test ($F(2, 46) = 12.0$, p < 0.001, $\eta_p^2 = 0.34$, *Figure 3B*) and the second test ($F(2, 46) = 14.8$, p < 0.001, $\eta_p^2 = 0.39$, *Figure 3C*). The happy BM induced a significantly larger pupil response than the neutral BM (first test: $t(23) = 2.60$, p = 0.022, Cohen's $d = 0.53$, 95% CI for the mean difference = [0.02, 0.14], Holm-corrected, p = 0.048 after Bonferroni correction, *Figure 3B*; second test: $t(23) = 3.36$, p = 0.005, Cohen's $d = 0.69$, 95% CI for the mean difference = [0.06, 0.24], Holm-corrected, p = 0.008

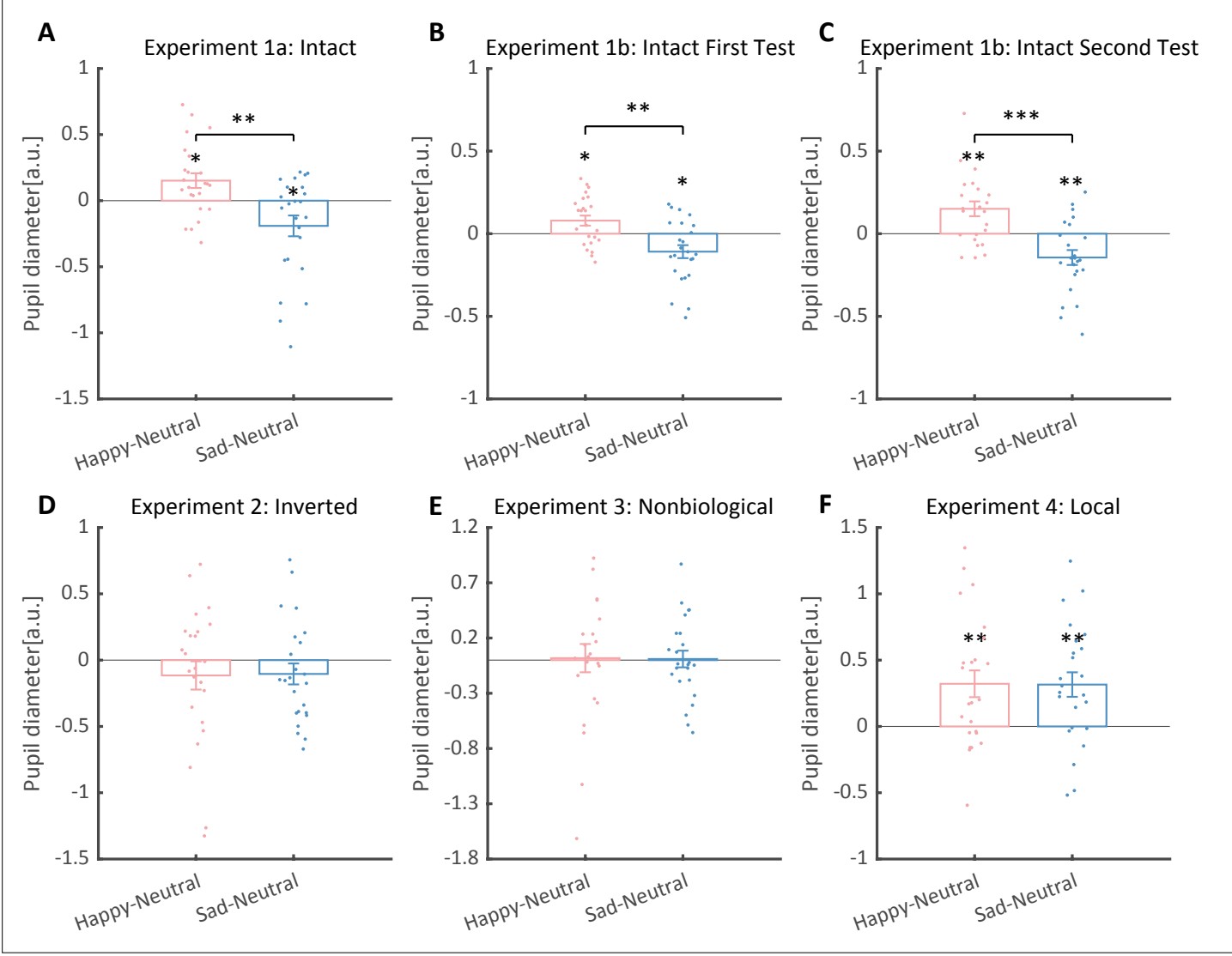

**Figure 3.** Normalized mean pupil responses using the neutral condition as baseline, plotted against happy and sad conditions. (**A**) In Experiment 1a, the group average pupil response to happy intact biological motion (BM) is significantly larger than that to sad and neutral BM, while the pupil size induced by sad BM is significantly smaller than that evoked by neutral BM. (**B, C**) Moreover, such an emotional modulation on pupil sizes was again identified in the test and retest of the replication experiment (Experiment 1b). (**D**) In Experiment 2, no significant differences in pupil responses were observed for inverted BM. (**E**) In Experiment 3, when the biological characteristic was deprived from the emotional BM, it failed to induce any modulations on pupil sizes. (**F**) In Experiment 4, both the happy and sad local BM induced a significantly larger pupil size than neutral local BM, with no significant difference between the happy and sad condition. All the pupil data are in arbitrary units (a.u.). Each point represents one individual data. Error bars showed standard errors of the mean (N = 24). *p < 0.05, **p < 0.01, ***p < 0.001.

after Bonferroni correction, *Figure 3C*). On the contrary, the sad BM induced a significantly smaller pupil response than the neutral BM (first test: $t(23)$ = −2.77, p = 0.022, Cohen's $d$ = 0.57, 95% CI for the mean difference = [−0.19, −0.03], Holm-corrected, p = 0.033 after Bonferroni correction, *Figure 3B*; second test: $t(23)$ = −3.19, p = 0.005, Cohen's $d$ = 0.65, 95% CI for the mean difference = [−0.24, −0.05], Holm-corrected, p = 0.012 after Bonferroni correction, *Figure 3C*). Besides, the happy BM induced significantly larger pupil response than the sad BM (first test: $t(23)$ = 4.23, p < 0.001, Cohen's $d$ = 0.86, 95% CI for the mean difference = [0.10, 0.28], Holm-corrected, p < 0.001 after Bonferroni correction, *Figure 3B*; second test: $t(23)$ = 4.26, p < 0.001, Cohen's $d$ = 0.87, 95% CI for the mean difference = [0.15, 0.44], Holm-corrected, p < 0.001 after Bonferroni correction, *Figure 3C*). The results of the cluster-based permutation analysis were also similar. The first test of Experiment 1b revealed that the happy BM induced significantly larger pupil responses than the neutral BM from

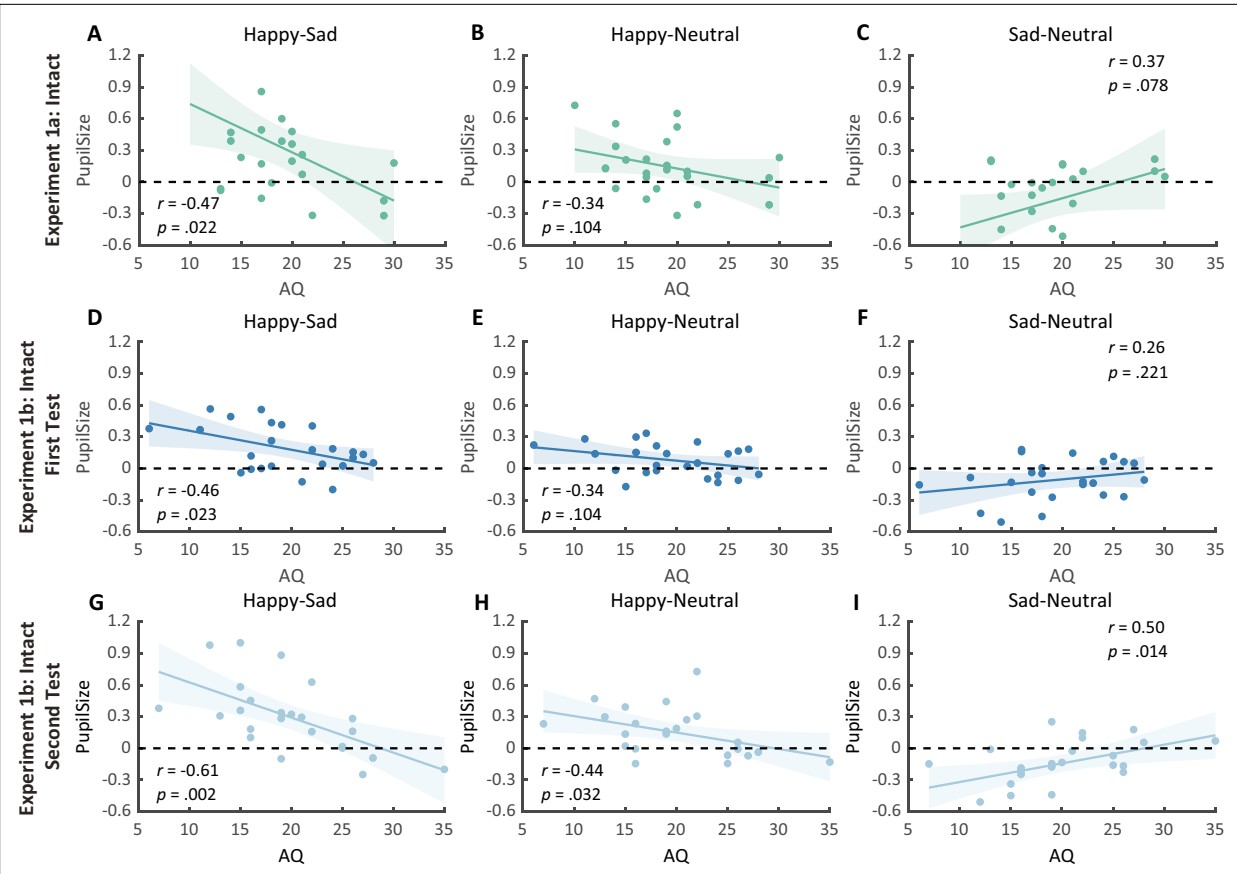

**Figure 4.** Correlation results for pupil modulations and autistic quotient (AQ) scores in Experiment 1a and its replication experiment (Experiment 1b). (**A**) In Experiment 1a, a significant negative correlation was found between the happy over sad pupil dilation effect and individual AQ. (**B, C**) No other significant correlations were found. (**D–F**) The first test of Experiment 1b replicated the negative correlation between happy over sad pupil dilation effect and AQ. Similarly, no other significant correlations were found. (**G**) In the second test, the negative correlation between happy over sad pupil dilation effect and AQ was similarly observed and even stronger. (**H, I**) Moreover, we have additionally observed a significant positive correlation between AQ and the happy minus neutral pupil dilation effect, and a significant negative correlation between AQ and the sad minus neutral pupil constriction effect. Dots in the scatter plot indicate the individual data and the shaded region indicates the 95% confidence interval (N = 24).

3250 to 4000 ms, and the sad BM evoked significantly smaller pupil response than the neutral BM from 2950 to 4000 ms. Additionally, the happy BM evoked a significantly larger pupil response as compared to the sad BM from 1450 to 4000 ms (see *Figure 6A*). Results of the second test revealed that the happy BM induced a significant pupil dilation effect than the neutral BM from 1900 to 4000 ms, and the sad BM evoked a significantly smaller pupil response than the neutral BM from 2450 to 4000 ms. Additionally, the happy BM evoked larger pupil responses as compared to the sad BM from from 1200 to 4000 ms (see *Figure 6B*).

Notably, we successfully replicated the negative correlation between the happy over sad dilation effect and individual autistic traits ($r(22) = -0.46$, $p = 0.023$, 95% CI for the mean difference = [−0.73, −0.07], *Figure 4D*) in the first test of Experiment 1b. No other significant correlations were found (see *Figures 4E, F* and *Figure 5B, C*). Moreover, in the second test, such a correlation was similarly found and was even stronger ($r(22) = -0.61$, $p = 0.002$, 95% CI for the mean difference = [−0.81, −0.27], *Figure 4G*). A test–retest reliability analysis was performed on the happy over sad pupil dilation effect and the AQ score, and the results showed robust correlations ($r$(happy–sad pupil size) = 0.56; $r$(AQ) = 0.90) and strong test–retest reliabilities ($\alpha$(happy–sad pupil size) = 0.60; $\alpha$(AQ) = 0.82). Furthermore, in the second test, we have additionally observed a significant negative correlation between AQ and the happy minus neutral pupil dilation effect ($r(22) = -0.44$, $p = 0.032$, 95% CI for the mean difference = [−0.72, −0.04], *Figure 4H*), and a significant positive correlation between the sad minus neutral pupil size and AQ ($r(22) = 0.50$, $p = 0.014$, 95% CI for the mean difference = [0.12, 0.75], *Figure 4I*). This

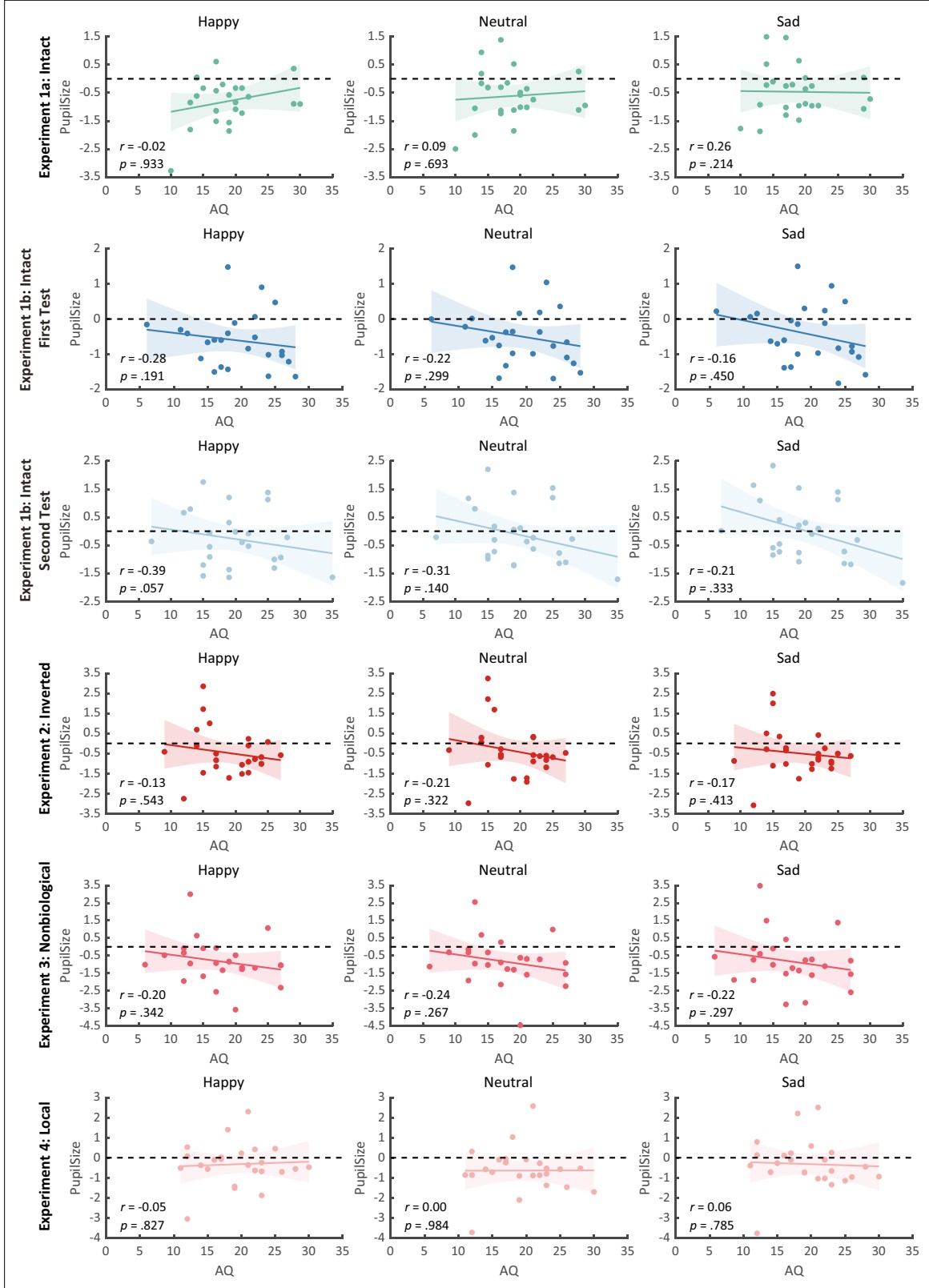

**Figure 5.** Correlation results for autistic quotient (AQ) scores and pupil responses toward happy, sad, and neutral biological motion (BM) across four experiments. No significant correlations were observed. Dots in the scatter plot indicate the individual data and the shaded region indicates the 95% confidence interval (N = 24).

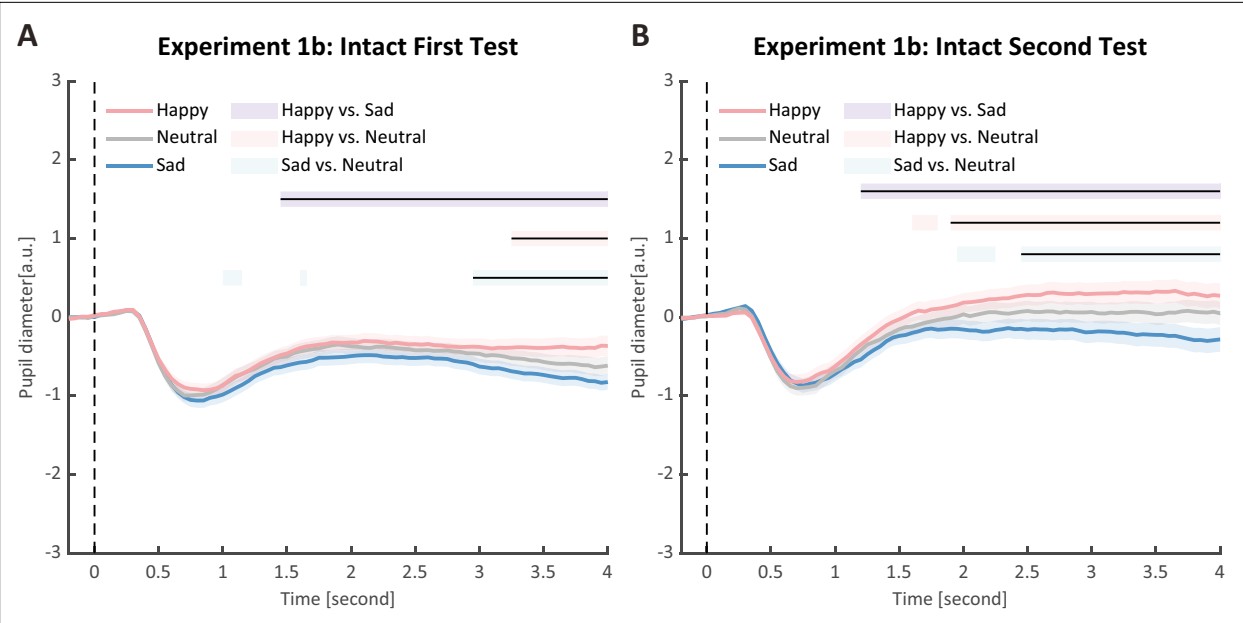

**Figure 6.** Time course of pupil responses to happy, sad, and neutral biological motion (BM) in Experiment 1b. Solid lines represent pupil diameter under each emotional condition as a function of time (happy: red; sad: blue; neutral: gray); shaded areas represent the standard error of the mean (SEM) between participants (N = 24); colored horizontal lines indicate periods during which there are statistically significant differences among conditions at p < 0.05; and black horizontal lines indicate significant differences after cluster-based permutation correction. All the pupil data are in arbitrary units (a.u.). (**A**) In the first test of Experiment 1b, we successfully replicated the results of Experiment 1a: the happy BM evoked larger pupil response as compared to the sad and neutral BM, and the sad BM evoked smaller pupil size than the neutral BM. (**B**) Such results were similarly observed in the retest.

indicated that the overall correlation between happy over sad dilation effect and AQ was the aggregate outcome of the diminished pupil modulations by happy and sad BM for high AQ individuals.

Overall, these results demonstrated a robust and replicable modulation of emotions embedded in the minimized point-light walker on pupil sizes: the happy BM evoked a larger pupil response than the neutral BM, while the sad BM evoked a smaller pupil size as compared to the neutral one. Importantly, a significant negative correlation between AQ and happy over sad pupil dilation effect was consistently found, and such effect was caused by a general attenuation in BM emotion perception sensitivity among individuals with high autistic tendencies.

## Experiment 2: inverted emotional BM

To rule out the possibility that the difference in low-level visual features rather than the emotional information per se might account for the obtained emotional modulation effect, we presented observers with the inverted BM stimuli that shared the exact perceptual features with their upright counterparts in Experiment 2. The cluster-based permutation analysis observed no significantly different time points among three emotional conditions (see *Figure 2B*). An identical one-way repeated measures ANOVA was conducted, while no significant main effect of emotional condition on pupil responses was observed in inverted BM ($F(2, 46) = 0.95$, $p = 0.396$, $\eta_p^2 = 0.04$, *Figure 3D*). Besides, no significant correlations between AQ and pupil modulations were found (AQ with happy–sad pupil size: $r(22) = 0.14$, $p = 0.512$, 95% CI for the mean difference = [−0.28, 0.52]; AQ with happy–neutral pupil size: $r(22) = 0.26$, $p = 0.227$, 95% CI for the mean difference = [−0.16, 0.60]; AQ with sad-neutral pupil size: $r(22) = 0.19$, $p = 0.376$, 95% CI for the mean difference = [−0.23, 0.55]). No other correlations were observed (*Figure 5D*). Critically, the mixed 2 (orientation: upright, inverted) × 3 (emotional condition: happy, sad, neutral) ANOVA on the average pupil size obtained in Experiments 1a and 2 showed a significant interaction between orientation and emotional condition ($F(2, 92) = 4.53$, $p = 0.013$, $\eta_p^2 = 0.09$), which could be attributed to the diminishment of emotional modulation in inverted BM. This indicated that the observed emotional modulation of pupil responses did not arise from low-level visual differences.

## Experiment 3: non-biological motion

Given the ample evidence showing that local motion feature is essential for the perception of biologically significant information from BM (*Chang and Troje, 2008*; *Chang and Troje, 2009*; *Troje and Westhoff, 2006*; *Wang and Jiang, 2012*), we moved forward to explore the role of local motion feature in the emotion processing of BM. In Experiment 3, we presented observers with non-BM stimuli, which were derived from the fragments identical to emotional BM but with critical local characteristics removed (*Chang and Troje, 2009*). The permutation analysis found no significantly different time points in pupil responses to the happy, sad, and neutral conditions (see *Figure 2D*). Results of the one-way repeated ANOVA on the average pupil sizes also showed no significant main effect of emotional condition ($F(1.6, 35.7) = 0.02$, p = 0.964, $\eta_p^2 = 0.00$; Greenhouse–Geisser corrected, see *Figure 3C*). Similarly, no pupil modulation effects significantly correlated with AQ (AQ with happy–sad pupil size: $r(22) = -0.03$, p = 0.896, 95% CI for the mean difference = [−0.43, 0.38]; AQ with happy–neutral pupil size: $r(22) = 0.01$, p = 0.956, 95% CI for the mean difference = [−0.39, 0.41]; AQ with sad–neutral pupil size: $r(22) = 0.06$, p = 0.777, 95% CI for the mean difference = [−0.35, 0.45]). No correlations were observed between AQ scores and pupil responses toward happy, sad, and neutral BM (*Figure 5E*). Moreover, combining the results obtained from Experiments 1a and 3, we found a significant interaction between stimulus type (intact, non-biological) and emotional condition (happy, sad, neutral) ($F(2, 92) = 3.22$, p = 0.045, $\eta_p^2 = 0.07$), which was caused by the lack of emotional modulation in non-biological stimuli. These findings together showed that the removal of local characteristics greatly disrupted the emotional modulations on observers' pupil responses, providing conceivable evidence for the critical contribution of local motion features in emotion perception from the BM signal.

## Experiment 4: local emotional BM

In Experiment 4, we went further to examine whether local BM alone could carry emotional information and exert a similar modulation effect on pupil size. In particular, we adopted the well-established local BM stimuli, the scrambled BM, whose local motion feature was retained while the global configuration information was completely disrupted. Both the explicit and implicit emotion processing of the scrambled BM were investigated to provide a thorough view of emotion perception from local BM. Specifically, a group of observers viewed the scrambled BM and made explicit behavioral judgments on the emotional information contained in the scrambled BM. The results showed that observers could successfully recognize the emotions contained in scrambled BMs with the average accuracy reaching significantly above 50% ($M$ ± standard error of mean [SEM] = 83 ± 2.9%, p < 0.001). This indicated that scrambled BM conveyed recognizable emotional information.

Furthermore, we investigated the pupil responses to scrambled happy, sad, and neutral BMs to explore the automatic and implicit processing of local emotional BM. The consecutive cluster-based permutation analysis further showed that both the scrambled happy and sad BM evoked larger pupil sizes than the scrambled neutral BM. Importantly, such effect appeared in a rather early time window and could last for a quite long time (happy vs. neutral: 450–3550 ms; sad vs. neutral: 350–4000 ms, see *Figure 2D*). The one-way repeated measures ANOVA on the mean pupil size indicated a significant main effect of emotional condition ($F(2, 46) = 6.47$, p = 0.003, $\eta_p^2 = 0.22$, *Figure 3F*). Follow-up analysis showed that the scrambled happy BM induced a significantly larger pupil response than the scrambled neutral BM ($t(23) = 3.22$, p = 0.008, Cohen's $d = 0.66$, 95% CI for the mean difference = [0.12, 0.53]; Holm-corrected, p = 0.011 after Bonferroni correction, *Figure 3F*), which is similar to that observed in Experiments 1a and 1b. Notably, the scrambled sad BM also induced a larger pupil size than the neutral one ($t(23) = 3.41$, p = 0.007, Cohen's $d = 0.70$, 95% CI for the mean difference = [0.12, 0.51]; Holm-corrected, p = 0.007 after Bonferroni correction, *Figure 3F*). Moreover, no difference in pupil size is observed between the happy and sad conditions ($t(23) = -0.07$, p = 0.948, Cohen's $d = 0.01$, 95% CI for the mean difference = [−0.23, 0.24]; Holm-corrected, p = 1.000 after Bonferroni correction, *Figure 3F*). The observed effect could not be accounted for by perceptual differences (e.g., speed), as the happy and sad scrambled BM differed the most in low-level features, whereas they induced similar dilation effects on pupil sizes. Again, no significant correlations were observed between AQ and pupil modulation effects (AQ with happy–sad pupil size: $r(22) = -0.21$, p = 0.32, 95% CI for the mean difference = [−0.57, 0.21]; AQ with happy–neutral pupil size: $r(22) = -0.13$, p = 0.56, 95% CI for the mean difference = [−0.50, 0.29]; AQ with sad–neutral pupil size: $r(22) = 0.13$, p = 0.56, 95% CI for the mean

difference = [−0.29, 0.50]). No other correlations were observed (*Figure 5F*). Importantly, a mixed 2 (BM type: intact, local) × 3 (emotional condition: happy, sad, neutral) ANOVA on average pupil size was conducted to examine whether or not this emotional modulation effect of local BM varied from that of intact BM reported in Experiments 1a. Results revealed a significant interaction between BM type and emotional condition ($F(2, 92) = 7.76$, p < 0.001, $\eta_p^2 = 0.14$), indicating that emotions in intact and local BM exerted differential modulations on pupil size. This interaction is primarily driven by the vanishment of pupil modulation between happy and sad local BM. Overall, these findings showed that the local BM could convey significant emotional information, which could induce a pupil dilation effect regardless of its exact type. As compared to the intact BM, the emotional modulation observed in local BM is different because it occurs faster and is rather coarse.

## Discussion

Life motion signals convey salient emotional information that is crucial for human survival and social interactions (*Johansson, 1973*; *Troje, 2008*). Here, we reported that such emotional clues carried by the point-light BM walker could exert a modulation effect on pupil responses. Specifically, the happy BM significantly dilated pupils as compared to the neutral BM, and the sad BM evoked smaller pupil responses than the neutral BM, showing distinct emotional modulations on pupil responses that depend on the emotional contents. Moreover, this emotional modulation of pupil responses could not be explained by the low-level differences, as viewing inverted BMs failed to induce such effects. Importantly, when the emotional BM was deprived of the local motion feature through the removal of accelerations, it failed to induce any modulation on pupil responses. Furthermore, the scrambled emotional BM with only the local motion feature retained could still produce a modulation effect on pupil size. Noticeably, this modulation is rapid but rather coarse: viewing the scrambled happy and sad BM would evoke greater pupil size than the scrambled neutral BM in a relatively early time window, while no significant difference was observed between the scrambled happy and sad BM. Taken together, these findings revealed multi-level processing of emotions in life motion signals: the global emotional BM-modulated pupil size in a fine-grained manner that discriminates each type of emotion, and the local emotional BM exerted a coarse but rapid modulation on pupil size.

The emotion processing of BM has been investigated in former studies using explicit behavioral detection and recognition paradigm. It has been reported that happy BM is more rapidly identified and detected as compared to sad and neutral BM (*Actis-Grosso et al., 2015*; *Lee and Kim, 2017*; *Spencer et al., 2016*), while other research observed no such superiority (*Chouchourelou et al., 2006*). Here, the present study adopted the novel and objective pupillometry index to investigate the emotion processing of BM from a physiological aspect. The pupil index, as a direct reflection of individual arousal level and cognitive state, served as a powerful tool for faithfully and automatically reflecting the implicit processing of emotions in BM. Previous studies have reported larger pupil responses toward emotional images as compared to neutral images, with no significant differences observed between the positive and negative conditions (*Bradley and Lang, 2015*; *Bradley et al., 2008*; *Snowden et al., 2016*). However, these studies mostly adopted complex emotional scene images that conveyed rather general emotional information. When it comes to the specific emotion cues (e.g., happy, sad) delivered by our conspecifics through biologically salient signals (e.g., faces, gestures, voices), the results became intermixed. Some studies reported pupil dilatory effects toward fearful, disgusted, angry, sad faces but not happy faces (*Burley et al., 2017*). On the contrary, other studies observed larger pupil responses for happy as compared to sad, fearful, and surprised faces (*Aktar et al., 2018*; *Burley and Daughters, 2020*; *Jessen et al., 2016*; *Prunty et al., 2022*). These conflicting results could be due to the low-level confounds of emotional faces (e.g., eye size) (*Carsten et al., 2019*; *Harrison et al., 2006*). Similar to faces, BM also conveyed salient clues concerning the emotional states of our interactive partners. However, they were highly simplified, and deprived of various irrelevant visual confounders (e.g., body shape). Here, we reported that the happy BM induced a stronger pupil response than the neutral and sad BM, lending support to the happy dilation effect observed with faces (*Burley and Daughters, 2020*; *Prunty et al., 2022*). Moreover, it helps ameliorate the concern regarding the low-level confounding factors by identifying similar pupil modulations in another type of social signal with distinctive perceptual features.

It has long been documented that pupil size reflects the degree of cognitive effort and attention input (*Joshi and Gold, 2020*; *van der Wel and van Steenbergen, 2018*), and indexes the noradrenalin

activity in emotion processing structures like amygdala (*Dal Monte et al., 2015*; *Harrison et al., 2006*; *Liddell et al., 2005*). Thus, the happy dilation effect suggests that the happy BM may be more effective in capturing attention and evoking emotional arousal than the neutral and sad BM, which echoes with the happiness superiority reported in the explicit processing of BM (*Lee and Kim, 2017*; *Spencer et al., 2016*; *Yuan et al., 2023*). Instead, the sad constriction effect potentially indicates a disadvantage for sad BM in attracting visual attention and evoking emotional arousal than neutral BM. In line with this, it has been found that infants looked more at the happy BM walker when displayed in pair with the neutral walker, whereas they attended less to the sad walker as compared to the neutral one (*Ogren et al., 2019*). Besides, it has been revealed by neural studies that, the happy emotion evoked stronger activities in emotionally relevant brain regions including the amygdala, the extra-striate body area, and the fusiform body area, while the sad emotion failed to induce such effects (*Peelen et al., 2007*; *Ross et al., 2019*). Still, future studies are needed to provide further evidence regarding the happy advantage and the sad disadvantage in attracting visual attention. For example, in addition to pupil size, the microsaccades also provided valuable insights into attention processes (*Baumeler et al., 2020*; *Engbert and Kliegl, 2003*; *Meyberg et al., 2017*), and potentially involved shared neural circuits with pupil responses (*Hafed et al., 2009*; *Hafed and Krauzlis, 2012*; *Wang et al., 2012*). Future studies could combine the microsaccade index with pupil size to provide a more thorough understanding of BM emotion processing.

Notably, the happy over sad dilation effect was negatively correlated with autistic traits: individuals with greater autistic tendencies showed decreased sensitivities to emotions in BM. In fact, the perception of BM has long been considered an important hallmark of social cognition, and abundant studies reported that individuals with social cognitive deficits (e.g., autism spectrum disorder, ASD) were impaired in BM perception (*Blake et al., 2003*; *Freitag et al., 2008*; *Klin et al., 2009*; *Nackaerts et al., 2012*). More recently, it has been pointed out that the extraction of more complex social information (e.g., emotions, intentions) from BM, as compared to basic BM recognitions, could be more effective in detecting ASDs (*Federici et al., 2020*; *Koldewyn et al., 2010*; *Parron et al., 2008*; *Todorova et al., 2019*). Specifically, a meta-analysis found that the effect size expanded nearly twice when the task required emotion recognition as compared to simple perception/detection (*Todorova et al., 2019*). However, for the high-functioning ASD individuals, it has been reported that they showed comparable performance with the control group in explicitly labeling BM emotions, while their responses were rather delayed (*Mazzoni et al., 2022*). This suggests that ASD individuals could adopt compensatory strategies to complete the explicit BM labeling task, while their automatic responses remained impaired. Such an observation highlights the importance of using more objective measures that do not rely on subjective reports to investigate the intrinsic perception of emotions from BM and its relationship with ASD-related social deficits. The current study thus introduced pupil size measurement to this field, and we combined it with the passive viewing task to investigate the more automatic aspect of BM emotion processing. In addition to diagnostic ASDs, the non-clinical general population also manifested autistic tendencies that followed normal distribution and demonstrated substantial heritability (*Hoekstra et al., 2007*). Here, we focused on the autistic tendencies in the general population, and our results showed that the automatic emotion processing of BM stimuli was impaired in individuals with high autistic tendencies, lending support to previous studies (*Hubert et al., 2007*; *Nackaerts et al., 2012*; *Parron et al., 2008*). The more detailed test–retest examination further confirmed such a correlation and illustrated a general diminishment in BM emotion perception ability (happy and sad) for high AQ individuals. Given that pupil measurement does not require any explicit verbal reports, it is easily attainable in children and even infants. Thus, the pupil modulation effect obtained here may serve as a sensitive and reliable physiological marker for detecting early social cognitive disorders in both clinical and non-clinical populations.

Remarkably, our findings highlighted the central role of local motion feature in modulating pupil responses toward emotional information contained in BM. Previous studies have shown that human visual system is highly sensitive to the local BM stimulus, whose global configural information is completely deprived through spatially scrambling (*Troje and Westhoff, 2006*). For example, it has been demonstrated that scrambled BM could perform as intact BM in lengthening subjective temporal perception (*Wang and Jiang, 2012*). Besides, such local motion feature also served as a basic pre-attentive feature in visual search (*Wang et al., 2010*) and could further induce a significant reflexive attentional effect (*Wang et al., 2014*; *Yu et al., 2020*). Moreover, the deprivation of the local

BM feature would greatly disrupt the perception of the contained biologically salient information, such as animacy and walking direction (*Chang and Troje, 2008*; *Chang and Troje, 2009*; *Yu et al., 2020*). Here, we extended this line of research to the emotional domain by demonstrating that the local BM component is also critical for the processing of emotional information: when the local BM feature is disrupted, the modulation of emotions on pupil size completely disappears. Furthermore, the emotional BM that retained only local motion features could still exert a salient modulation effect on pupil size. In particular, the happy and sad local BM induced a significant pupil dilation effect as compared to the neutral local BM stimuli. Intriguingly, this dilation effect is independent of the specific emotion category but reflects a general activation caused by the affective salient information. In addition, this non-selective pupil dilation effect in local BM appeared in a relatively early time window, indicating that the extraction of emotions from local BM is rapid. Taken together, these findings identified a coarse but rapid emotion processing mechanism in local BM that could promptly detect the emotional information therein without the aid of the global shape.

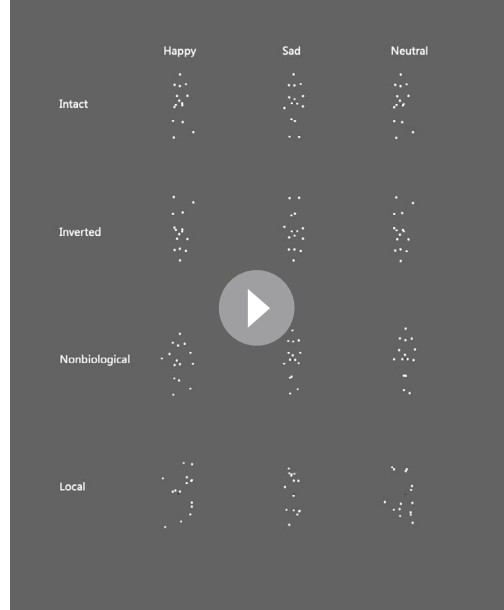

**Video 1.** Demonstration of motion stimuli used in Experiments 1–4.
https://elifesciences.org/articles/89873/figures#video1

Moreover, our findings revealed the existence of distinct emotional modulations in intact and local BM, respectively, as evidenced by variations in pupil responses. In particular, the happy intact BM induced a significant pupil dilation effect as compared to the neutral one, whereas the sad intact BM led to pupil constriction. Interestingly, both the happy and sad local BM resulted in pupil dilation effects relative to the neutral local BM, and such effect occurred at a relatively early time point. This distinctive pupil modulation effects observed in intact and local BM could be elucidated by a multi-level emotion processing mechanism. Specifically, even though both the intact and local BM conveyed important life information, the local BM is deprived of the global configural information (*Chang and Troje, 2008*; *Chang and Troje, 2009*; *Simion et al., 2008*). Thus, its emotion processing potentially occurred at a more basic and preliminary level, responding to the general affective salient information without engaging in further detailed analysis. Importantly, similar dissociated emotion processing phenomenon has been observed in another important type of emotional signal with analogous function (i.e., facial expression). For example, happy and fearful faces induced differential amygdala activations when perceived consciously, yet they induced comparable amygdala activations when suppressed (*Williams et al., 2004*). Moreover, it has been argued that there exist two parallel pathways for facial expression processing: a slow route that conveys fine-grained facial features along cortical areas to the amygdala, and a rapid subcortical pathway that directly transfers emotional information to amygdala without detailed analysis, which is known as the 'quick and dirty' route (*Garrido et al., 2012*; *Johnson, 2005*; *Méndez-Bértolo et al., 2016*; *Vuilleumier et al., 2003*). It is probable that the emotion processing of the local and intact point-light BM potentially functions in a manner similar to that of the faces, with the former serving as a primary detection mechanism that automatically captures emotionally significant information conveyed by animate agents without detailed analysis, and the latter aiding in more specific emotion identification based on fine-grained analyses of their motion pattern and global shape. Compatible with this view, recent studies have reported that the emotion processing of face and BM was very similar. For example, they both showed a happiness superiority in visual detection (*Becker et al., 2011*; *Lee and Kim, 2017*; *Nackaerts et al., 2012*) and guiding social attention (*Yuan et al., 2023*). Moreover, their emotion processing involved potentially overlapping brain regions, such as amygdala, superior temporal sulcus (STS) (*Alaerts et al., 2014*; *Engell and Haxby, 2007*; *Peelen et al., 2007*; *Pessoa and Adolphs, 2010*). Overall,

our findings, together with the former evidence, suggest *Video 1* dissociated emotion processing for BM in the human brain akin to the dual-route model reported in faces, and further imply a general 'emotional brain' that can be shared by different types of social signals. Still, future studies are needed to implement neuroimaging techniques to directly identify the brain regions involved in the emotion processing of local and global BM signals.

To conclude, the current study clearly demonstrated that the emotional information conveyed by point-light BM-modulated pupil responses. The intact BM exerted a fine-grained emotional modulation on pupil sizes, while disrupting the contained local motion characteristic would deteriorate the observed modulations. Moreover, BM with only local motion features retained could exert a fast but rather coarse modulation on pupillometry. These findings together highlight the critical role of local motion feature in BM emotion processing, and further reveal the multi-level emotion processing of BM. Notably, the observed emotional modulation in intact BM is associated with individual autistic traits, suggesting the potential of utilizing the emotion-modulated pupil responses to facilitate the diagnosis of social cognitive disorders.

## Methods

### Participants

A total of 144 participants (50 males, 94 females) ranging from 18 to 30 years old ($M \pm$ standard deviation [SD] = 23.1 ± 2.5) were recruited, with 24 in Experiment 1a, 24 in Experiment 1b, 24 in Experiment 2, 24 in Experiment 3, 24 in the behavioral part of Experiment 4, and 24 in the eye recoding part of Experiment 4. All of the participants had normal or corrected-to-normal vision and gave written informed consent in accordance with the procedure and protocols approved by the institutional review board of the Institute of Psychology, Chinese Academy of Sciences. They were all naive to the purpose of the experiments. Prior power analyses ($F$ tests, repeated measures, within factors) using G*Power (Version 3.1.9.4; *Faul et al., 2007*) indicated that a sample size of 24 participants would afford 80% power with alpha at 0.05 to detect a moderate main effect ($f$ = 0.27). This sample size was comparable to previous studies with similar designs (*Nakakoga et al., 2020*). We report how we determined our sample size, all data exclusions (if any), all manipulations, and all measures following JARS (*Kazak, 2018*).

### Stimuli

Stimuli were displayed using MATLAB (Mathworks, Inc) together with the Psychtoolbox extensions (*Brainard, 1997*; *Pelli, 1997*) on a 23-inch LED monitor (1920 × 1080 at 60 Hz) with gray background (red-green-blue [RGB]: 100, 100, 100). The BM stimuli were adopted from *Troje, 2008*. Each comprised 15 point-light dots depicting the motions of the head, pelvis, thorax, and major joints (i.e., shoulders, elbows, wrists, hips, knees, and ankles). Its emotional state was indexed by a normalized $Z$ score on an axis that reflects the differences between happy and sad walkers in terms of a linear classifier. The scores were computed within a 10-dimensional sub-space spanned by the first 10 principal components based on a Fourier-based representation of observers' emotional ratings of 80 actual walkers (half male) (*Troje, 2008*). We adopted the neutral walker that scored 0 on the linear axis, together with the happy walker 6 SDs into the happy part of the axis and the sad walker 6 SDs into the sad part of the axis (see https://www.biomotionlab.ca/html5-bml-walker/ for an interactive animation). Besides, we turned the BM walkers 45° leftwards or rightwards to maximize the visibility of expressive bodily cues (*Roether et al., 2009*). In Experiments 1a and 1b, the upright emotional (happy, sad, or neutral) BM walkers with two walking directions (45° leftwards or rightwards) were used. The velocity was 5.76 pixels/frame for the happy BM, 4.14 pixels/frame for the neutral BM, and 3.21 pixels/frame for the sad BM. This difference in velocity profile was considered an important signature for conveying emotional information, as the happy walker was characterized by a larger step pace and longer arm swing, and the sad walker would instead exhibit a slouching gait with short slow strides and smaller arm movement (*Barliya et al., 2013*; *Chouchourelou et al., 2006*; *Halovic and Kroos, 2018*; *Roether et al., 2009*). In Experiment 2, the stimuli were replaced with their inverted counterparts created by mirror flipping the upright BM vertically. In Experiment 3, we presented observers with the non-BM stimuli whose local BM characteristic was disrupted through acceleration removal. More specifically, each individual dot moved along the original path with a constant speed equal to the average speed

of the dot (*Chang and Troje, 2009*). Such manipulation disrupted the local motion feature of the BM stimuli but kept the trajectories of individual point lights unchanged. In Experiment 4, observers viewed the scrambled BM stimuli, which were created by randomly relocating the point-light dots within the region occupied by the original BM walker. In this manner, the local motion feature was preserved while the global configuration of the BM stimulus was entirely disrupted (*Troje and Westhoff, 2006*) (see *Video 1* for stimuli demonstration).

## Procedure

Participants were seated at a viewing distance of 60 cm to the computer screen with their heads on a chinrest to minimize their head movements. Their pupil size and eye position of the left eye were recorded using a video-based iView X Hi-Speed system (SMI, Berlin, Germany) at 500 Hz. In Experiment 1a, each trial began with a central fixation cross (0.2° × 0.2°) with variable duration (800–1200 ms), followed by an upright happy/sad/neutral point-light walker (half leftwards and half rightwards) presented centrally for 4000 ms. Participants were required to passively view the BM stimulus and continue the procedure by pressing the space bar. After that, a blank screen was displayed for 3000 ms (*Figure 1*). There were 4 blocks of 30 trials, and participants were given a short break after every block. Besides, we also administered a 9-point calibration of the eye-tracker followed by a validation stage before each experimental block to ensure the data quality. Participants were told to maintain fixation on the center of the screen and not to blink during the stimuli presentation. Importantly, a replication experiment of Experiment 1a (Experiment 1b) was conducted, which followed the same protocol, with the change being that participants were asked to return to the lab for a retest after at least 7 days. Experiment 2 is identical to Experiment 1a, except that the BM stimuli were changed to their inverted counterparts. In Experiment 3, we instead presented participants with the non-BM displays whose local motion feature was deprived through acceleration removal. In Experiment 4, we adopted the scrambled emotional BM stimuli, and investigated both the explicit and the implicit emotion processing of scrambled BM. Note that a group of participants was recruited to perform emotional judgments on the scrambled BM in order to investigate whether the emotional information carried by local BM signals could be explicitly identified. Another group of participants was enrolled for the pupil recording experiment, which followed the identical experimental procedure of Experiments 1–3. At the end of all experiments, participants were required to complete the 50-point autism-spectrum questionnaire (AQ), which measured the degree of autistic traits in the normal population (*Baron-Cohen et al., 2001*).

## Data analysis

The raw pupil data for each trial were first screened to remove eye blinks (either replaced by linear interpolation or with this trial discarded). Then, trials with pupil size deviating ±3 SDs from the mean were excluded from further analysis. Finally, the pupil size data were down-sampled to 20 Hz and baseline-corrected for each trial by subtracting the mean pupil size during the 200 ms pre-stimulus period. We computed the average pupil size for each emotional condition (happy, sad, or neutral) obtained by collapsing the pupillometry across all time points. Besides, to depict the time course of pupil responses toward emotional BM, we further conducted a consecutive paired-sample *t*-test across all time points comparing different emotional conditions. The cluster-based permutation analysis was applied to avoid potential problems brought about by multiple comparisons (*Einhäuser et al., 2008*). In this analysis, the computed *t*-values neighboring in time that exceeded a threshold (p < 0.05) were defined as clusters, and then summed to produce a cluster mass. The cluster mass was compared with a null distribution, which was generated by 2000 random permutations of the pupil data from different conditions. If the cluster mass fell beyond 95% of the null distribution ($\alpha = 0.05$), it was deemed to be statistically significantly different (*Einhäuser et al., 2008*). The pupil size was analyzed and reported in arbitrary units without transforming into the actual unit (mm), as the relative change of the pupil size was of main interest.

## Acknowledgements

This research was supported by grants from STI2030-Major Projects (No. 2021ZD0203800 and 2022ZD0205100), the National Natural Science Foundation of China (No. 32430043 and 32371106), the Interdisciplinary Innovation Team of the Chinese Academy of Sciences (JCTD-2021-06), the Key

Research and Development Program of Guangdong, China (2023B0303010004), and the Fundamental Research Funds for the Central Universities. We thank Professor Nikolaus F Troje for kindly providing us with the visual stimuli.

## Additional information

### Funding

| Funder | Grant reference number | Author |
|---|---|---|
| Ministry of Science and Technology of the People's Republic of China | 2021ZD0203800 | Yi Jiang |
| Ministry of Science and Technology of the People's Republic of China | 2022ZD0205100 | Li Wang |
| National Natural Science Foundation of China | 32430043 | Yi Jiang |
| Interdisciplinary Innovation Team | JCTD-2021-06 | Yi Jiang |
| Fundamental Research Funds for Central Universities | | Yi Jiang |
| National Natural Science Foundation of China | 32371106 | Li Wang |
| Key Research and Development Program of Guangdong, China | 2023B0303010004 | Yi Jiang |

The funders had no role in study design, data collection, and interpretation, or the decision to submit the work for publication.

### Author contributions

Tian Yuan, Conceptualization, Data curation, Software, Formal analysis, Investigation, Visualization, Methodology, Writing - original draft; Li Wang, Yi Jiang, Conceptualization, Resources, Supervision, Funding acquisition, Investigation, Methodology, Writing - review and editing

### Author ORCIDs

Tian Yuan ⓘ https://orcid.org/0000-0002-8570-1484
Li Wang ⓘ https://orcid.org/0000-0002-2204-5192
Yi Jiang ⓘ https://orcid.org/0000-0002-5746-7301

### Ethics

All procedures contributing to this work comply with the ethical standards of the relevant national and institutional committees on human experiments and with the Helsinki Declaration of 1975, as revised in 2008. Written informed consent, and consent to publish, was obtained from participants. The institutional review board of the Institute of Psychology, Chinese Academy of Sciences has approved this study (reference number for approval: H18029).

Reviewer #1 (Public review): https://doi.org/10.7554/eLife.89873.3.sa1
Reviewer #2 (Public review): https://doi.org/10.7554/eLife.89873.3.sa2
Reviewer #3 (Public review): https://doi.org/10.7554/eLife.89873.3.sa3
Author response https://doi.org/10.7554/eLife.89873.3.sa4

## Additional files

### Supplementary files
• MDAR checklist

### Data availability
All data, materials, and analysis code used in the current study could be accessed at Science Data Bank (https://doi.org/10.57760/sciencedb.psych.00125).

The following dataset was generated:

| Author(s) | Year | Dataset title | Dataset URL | Database and Identifier |
|---|---|---|---|---|
| Yuan T, Li Wang, Jiang Y | 2024 | Data from: Multi-level processing of emotions in life motion signals revealed through pupil responses | https://doi.org/10.57760/sciencedb.psych.00125 | Science Data Bank, 10.57760/sciencedb.psych.00125 |

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
