## [Editor Report · eLife Assessment]

This **important** study provides **convincing** evidence that emotional information in biological motion can induce different patterns of pupil responses, which could serve as a behavioral marker of an autistic trait. These results broaden our understanding of how emotional biological motion can automatically trigger physiological changes and reveal the potential of using emotional-modulated pupil response to facilitate the diagnosis of social cognitive disorders. The work will be of broad interest to cognitive neuroscience, psychology, affective neuroscience, and vision science.

---

## [Referee Report · Reviewer #1 (Public review)]

Summary:

Tian et al. investigated the effects of emotional signals in biological motion on pupil responses. In this study, subjects were presented with point-light biological motion stimuli with happy, neutral, and sad emotions. Their pupil responses were recorded with an eye tracker. Throughout the study, emotion type (i.e., happy/sad/neutral) and BM stimulus type (intact/inverted/non-BM/local) were systematically manipulated. For intact BM stimuli, happy BM induced a larger pupil diameter than neutral BM, and neutral BM also induced a larger pupil diameter than sad BM. Importantly, the diameter difference between happy and sad BM correlated with the autistic trait of individuals. These effects disappeared for the inverted BM and non-BM stimuli. Interestingly, both happy and sad emotions show superiority in pupil diameter.

Strengths:

(1) The experimental conditions and results are very easy to understand.

(2) The writing and data presentation are clear.

(3) The methods are sound. I have no problems with the experimental design and results.

---

## [Referee Report · Reviewer #2 (Public review)]

Summary:

Through a serial of four experiments, Yuan, Wang and Jiang examined pupil size responses to emotion signals in point-light motion stimuli. Experiment 1 examined upright happy, sad and neutral point-light biological motion (BM) walkers. The happy BM induced a significantly larger pupil response than the neutral, whereas the sad BM evoked a significantly smaller pupil size than the neutral BM. Experiment 2 examined inverted BM walkers. Experiment 3 examined BM stimuli with acceleration removed. No significant effects of emotion were found in neither Experiment 2 nor Experiment 3. Experiment 4 examined scrambled BM stimuli, in which local motion features were preserved while the global configuration was disrupted. Interestingly, the scrambled happy and sad BM led to significant greater pupil size than the scrambled neutral BM at a relatively early time, while no significant difference between the scrambled happy and sad BM was found. Thus, the authors argue that these results suggest multi-level processing of emotions in life motion signals.

Strengths:

The experiments were carefully designed and well-executed, with point-light stimuli that eliminate many potential confounding effects of low-level visual features such as luminance, contrast, and spatial frequency.

Overall, I think this is a well-written paper with solid experimental results that support the claim of the authors, i.e., the human visual system may process emotional information in biological motion at multiple levels. Given the key role of emotion processing in normal social cognition, the results will be of interest not only to basic scientists who study visual perception, but also to clinical researchers who work with patients of social cognitive disorders. In addition, this paper suggests that examining pupil size responses could be a very useful methodological tool to study brain mechanisms underlying emotion processing.

---

## [Referee Report · Reviewer #3 (Public review)]

Summary:

The overarching goal of the authors was to understand whether emotional information conveyed through point-light biological motion can trigger automatic physiological responses, as reflected in pupil size.

Strengths:

This manuscript has several noticeable strengths: it addresses an intriguing research question that fills that gap in existing literature, presents a clear and accurate presentation of the current literature, and conducts a series of experiments and control experiments with adequete sample size. Yet, it also entails several noticeable limitations - especially in the study design and statistical analyses.

Assessment of the revision:

The authors have done a thorough job revising the manuscript, effectively addressing all of my previous concerns.

---

## [Author Response]

The following is the authors’ response to the original reviews.

**eLife assessment:**
This study presents an important finding on the implicit and automatic emotion perception from biological motion (BM). The evidence supporting the claims of the authors is solid, although inclusion of a larger number of samples and more evidence for the discrepancy between Intact and local emotional BMs would have strengthened the study. The work will be of broad interest to perceptual and cognitive neuroscience.

We express our sincere gratitude for the positive and constructive evaluation of our manuscript. We have now included more participants and conducted a replication experiment to strengthen our results.

**Reviewer #1 (Public Review):**
Summary:Tian et al. investigated the effects of emotional signals in biological motion on pupil responses. In this study, subjects were presented with point-light biological motion stimuli with happy, neutral, and sad emotions. Their pupil responses were recorded with an eye tracker. Throughout the study, emotion type (i.e., happy/sad/neutral) and BM stimulus type (intact/inverted/non-BM/local) were systematically manipulated. For intact BM stimuli, happy BM induced a larger pupil diameter than neutral BM, and neutral BM also induced a larger pupil diameter than sad BM. Importantly, the diameter difference between happy and sad BM correlated with the autistic trait of individuals. These effects disappeared for the inverted BM and non-BM stimuli. Interestingly, both happy and sad emotions show superiority in pupil diameter.Strengths:(1) The experimental conditions and results are very easy to understand.(2) The writing and data presentation are clear.(3) The methods are sound. I have no problems with the experimental design and results.Weaknesses:(1) My main concern is the interpretation of the intact and local condition results. The processing advantage of happy emotion is not surprising given a number of existing studies. However, the only difference here seems to be the smaller (or larger) pupil diameter for sad compared to neutral in the intact (or local, respectively) condition. The current form only reports this effect but lacks in-depth discussions and explanations as to why this is the case.

Thanks for pointing this out, our apology for not making this point clear. It has long been documented that pupil size reflects the degree of cognitive effort and attention input (Joshi & Gold, 2019; van der Wel & van Steenbergen, 2018), and indexes the noradrenalin activity in emotion processing structures like amygdala (Dal Monte et al., 2015; Harrison et al., 2006; Liddell et al., 2005). Accordingly, we proposed that the smaller pupil response observed under the sad condition as compared to the neutral condition is because the sad biological motion (BM) could be less efficient in attracting visual attention and evoking emotional arousal. In line with this, it has been found that infants looked more at the neutral point-light walker when displayed in pair with the sad walker (Ogren et al., 2019), suggesting that the sad BM is less effective in capturing visual attention than the neutral BM. Besides, neural studies have revealed that, compared with other emotions (anger, happiness, disgust, and fear), the processing of sad emotion failed to evoke heightened activities in any emotionally relevant brain regions including the amygdala, the extrastriate body area (EBA) and the fusiform body area (FBA) (Peelen et al., 2007)(Peelen et al., 2007). The current study echoed with these previous findings by demonstrating a disadvantage for intact sad BM in evoking pupil responses. Notably, different from the intact sad BM, the local sad BM would instead induce stronger pupil responses than the neutral local BM. This distinctive pupil modulation effect observed in intact and local sad BM could be explained as a multi-level emotion processing model of BM. Specifically, even though both the intact and local BM conveyed important life information (Chang & Troje, 2008, 2009; Simion et al., 2008), the latter is deprived of the global form feature. Hence, the processing of emotions in local BM may occur at a more basic and preliminary level, responding to the general affective salient emotion information (happy and sad) without detailed analysis. In fact, similar dissociated emotion processing phenomenon has been observed in another important type of emotional signal with analogous function (i.e., facial expression). For example, happy and fearful faces elicited differential amygdala activations when perceived consciously. However, they elicited comparable amygdala activations when suppressed (Williams et al., 2004). Moreover, it has been proposed that there exist two parallel routes for facial expression processing: a quick but coarse subcortical route that detects affective salient information without detailed analysis, and a fine-grained but slow cortical route that discriminates the exact emotion type. Similarly, the dissociated emotion processing in local and intact BM may function in the same manner, with the former serving as a primary emotion detection mechanism and the latter serving as a detailed emotion discrimination mechanism. Still, future studies adopting more diverse experimental paradigms and neuroimaging techniques were needed to further investigate this issue. We have added these points and more thoroughly discussed the potential mechanism in the revised text (see lines 329-339, 405-415, 418-420).

References:

Chang, D. H. F., & Troje, N. F. (2008). Perception of animacy and direction from local biological motion signals. Journal of Vision, 8(5), 3. https://doi.org/10.1167/8.5.3

Chang, D. H. F., & Troje, N. F. (2009). Characterizing global and local mechanisms in biological motion perception. Journal of Vision, 9(5), 8–8. https://doi.org/10.1167/9.5.8

Dal Monte, O., Costa, V. D., Noble, P. L., Murray, E. A., & Averbeck, B. B. (2015). Amygdala lesions in rhesus macaques decrease attention to threat. Nature Communications, 6(1). https://doi.org/10.1038/ncomms10161

Harrison, N. A., Singer, T., Rotshtein, P., Dolan, R. J., & Critchley, H. D. (2006). Pupillary contagion: central mechanisms engaged in sadness processing. Social Cognitive and Affective Neuroscience, 1(1), 5–17. https://doi.org/10.1093/scan/nsl006

Joshi, S., & Gold, J. I. (2019). Pupil size as a window on neural substrates of cognition. Trends in Cognitive Sciences, 24(6), 466–480. https://doi.org/10.31234/osf.io/dvsme

Liddell, B. J., Brown, K. J., Kemp, A. H., Barton, M. J., Das, P., Peduto, A., Gordon, E., & Williams, L. M. (2005). A direct brainstem–amygdala–cortical ‘alarm’ system for subliminal signals of fear. NeuroImage, 24(1), 235–243.

Ogren, M., Kaplan, B., Peng, Y., Johnson, K. L., & Johnson, S. P. (2019). Motion or emotion: infants discriminate emotional biological motion based on low-level visual information. Infant Behavior and Development, 57, 101324. https://doi.org/10.1016/j.infbeh.2019.04.006

Peelen, M. V., Atkinson, A. P., Andersson, F., & Vuilleumier, P. (2007). Emotional modulation of body-selective visual areas. Social Cognitive and Affective Neuroscience, 2(4), 274–283. https://doi.org/10.1093/scan/nsm023

Simion, F., Regolin, L., & Bulf, H. (2008). A predisposition for biological motion in the newborn baby. Proceedings of the National Academy of Sciences, 105(2), 809–813. https://doi.org/10.1073/pnas.0707021105

van der Wel, P., & van Steenbergen, H. (2018). Pupil dilation as an index of effort in cognitive control tasks: a review. Psychonomic Bulletin & Review, 25(6), 2005–2015. https://doi.org/10.3758/s13423-018-1432-y

Williams, M. A., Morris, A. P., McGlone, F., Abbott, D. F., & Mattingley, J. B. (2004). Amygdala responses to fearful and happy facial expressions under conditions of binocular suppression. Journal of Neuroscience, 24(12), 2898-2904.

(2) I also found no systematic discussion and theoretical contributions regarding the correlation with the autistic traits. If the main point of this paper is to highlight an implicit and objective behavioral marker of the autistic trait, more interpretation and discussion of the links between the results and existing findings in ASD are needed.

We thank the reviewer for this insightful suggestion. The perception of biological motion (BM) has long been considered an important hallmark of social cognition. Abundant studies reported that individuals with social cognitive deficits (e.g., ASD) were impaired in BM perception (Blake et al., 2003; Freitag et al., 2008; Klin et al., 2009; Nackaerts et al., 2012). More recently, it has been pointed out that the extraction of more complex social information (e.g., emotions, intentions) from BM, as compared to basic BM recognitions, could be more effective in detecting ASDs (Federici et al., 2020; Koldewyn et al., 2009; Parron et al., 2008; Todorova et al., 2019). Specifically, a meta-analysis found that the effect size expanded nearly twice when the task required emotion recognition as compared to simple perception/detection (Todorova et al., 2019). However, for the high-functioning ASD individuals, it has been reported that they showed comparable performance with the control group in explicitly labelling BM emotions, while their responses were rather delayed (Mazzoni et al., 2021). This suggested that ASD individuals could adopt compensatory strategies to complete the explicit BM labelling task, while their automatic behavioural responses remained impaired. This highlights the importance of using more objective measures that do not rely on active reports to investigate the intrinsic perception of emotions from BM and its relationship with ASD-related social deficits. The current study thus introduced the pupil size measurement to this field, and we combined it with the passive viewing task to investigate the more automatic aspect of BM emotion processing. More importantly, in addition to diagnostic ASDs, the non-clinical general population also manifested autistic tendencies that followed normal distribution and demonstrated substantial heritability (Hoekstra et al., 2007). Here, we focused on the autistic tendencies in the general population, and our results showed that pupil modulations by BM emotions were indicative of individual autistic traits. Specifically, passively viewing the happy BMs evoked larger pupil responses than the sad BMs, while such emotional modulation diminished with the increase of autistic tendencies. More detailed test-retest examination further illustrated such a correlation was driven by the general diminishment in pupil modulation effects by emotional BM (happy or sad) for individuals with high autistic tendencies. This finding demonstrated that the automatic emotion processing of BM stimuli was impaired in individuals with high autistic tendencies, lending support to previous studies (Hubert et al., 2006; Nackaerts et al., 2012; Parron et al., 2008). This indicated the utility of emotional BM stimuli and pupil measurement in identifying ASD-related tendencies in both clinical and non-clinical populations. We have added these points to the revised text (see lines 347-375).

References:

Blake, R., Turner, L. M., Smoski, M. J., Pozdol, S. L., & Stone, W. L. (2003). Visual recognition of biological motion is impaired in children with autism. Psychological Science, 14(2), 151–157. https://doi.org/10.1111/1467-9280.01434

Federici, A., Parma, V., Vicovaro, M., Radassao, L., Casartelli, L., & Ronconi, L. (2020). Anomalous perception of biological motion in autism: a conceptual review and meta-analysis. Scientific Reports, 10(1). https://doi.org/10.1038/s41598-020-61252-3

Freitag, C. M., Konrad, C., Häberlen, M., Kleser, C., von Gontard, A., Reith, W., Troje, N. F., & Krick, C. (2008). Perception of biological motion in autism spectrum disorders. Neuropsychologia, 46(5), 1480–1494. https://doi.org/10.1016/j.neuropsychologia.2007.12.025

Hoekstra, R. A., Bartels, M., Verweij, C. J. H., & Boomsma, D. I. (2007). Heritability of autistic traits in the general population. Archives of Pediatrics & Adolescent Medicine, 161(4), 372. https://doi.org/10.1001/archpedi.161.4.372

Hubert, B., Wicker, B., Moore, D. G., Monfardini, E., Duverger, H., Fonséca, D. D., & Deruelle, C. (2006). Brief report: recognition of emotional and non-emotional biological motion in individuals with autistic spectrum disorders. Journal of Autism and Developmental Disorders, 37(7), 1386–1392. https://doi.org/10.1007/s10803-006-0275-y

Klin, A., Lin, D. J., Gorrindo, P., Ramsay, G., & Jones, W. (2009). Two-year-olds with autism orient to non-social contingencies rather than biological motion. Nature, 459(7244), 257–261. https://doi.org/10.1038/nature07868

Koldewyn, K., Whitney, D., & Rivera, S. M. (2009). The psychophysics of visual motion and global form processing in autism. Brain, 133(2), 599–610. https://doi.org/10.1093/brain/awp272

Mazzoni, N., Ricciardelli, P., Actis-Grosso, R., & Venuti, P. (2021). Difficulties in recognising dynamic but not static emotional body movements in autism spectrum disorder. Journal of Autism and Developmental Disorders, 52(3), 1092–1105. https://doi.org/10.1007/s10803-021-05015-7

Nackaerts, E., Wagemans, J., Helsen, W., Swinnen, S. P., Wenderoth, N., & Alaerts, K. (2012). Recognizing biological motion and emotions from point-light displays in autism spectrum disorders. PLoS ONE, 7(9), e44473. https://doi.org/10.1371/journal.pone.0044473

Parron, C., Da Fonseca, D., Santos, A., Moore, D. G., Monfardini, E., & Deruelle, C. (2008). Recognition of biological motion in children with autistic spectrum disorders. Autism, 12(3), 261–274. https://doi.org/10.1177/1362361307089520

Todorova, G. K., Hatton, R. E. M., & Pollick, F. E. (2019). Biological motion perception in autism spectrum disorder: a meta-analysis. Molecular Autism, 10(1). https://doi.org/10.1186/s13229-019-0299-8

**Reviewer #2 (Public Review):**
Summary:Through a series of four experiments, Yuan, Wang and Jiang examined pupil size responses to emotion signals in point-light motion stimuli. Experiment 1 examined upright happy, sad and neutral point-light biological motion (BM) walkers. The happy BM induced a significantly larger pupil response than the neutral, whereas the sad BM evoked a significantly smaller pupil size than the neutral BM. Experiment 2 examined inverted BM walkers. Experiment 3 examined BM stimuli with acceleration removed. No significant effects of emotion were found in neither Experiment 2 nor Experiment 3. Experiment 4 examined scrambled BM stimuli, in which local motion features were preserved while the global configuration was disrupted. Interestingly, the scrambled happy and sad BM led to significantly greater pupil size than the scrambled neutral BM at a relatively early time, while no significant difference between the scrambled happy and sad BM was found. Thus, the authors argue that these results suggest multi-level processing of emotions in life motion signals.Strengths:The experiments were carefully designed and well-executed, with point-light stimuli that eliminate many potential confounding effects of low-level visual features such as luminance, contrast, and spatial frequency.Weaknesses:Correlation results with limited sample size should be interpreted with extra caution.

Thanks for pointing this out. To strengthen the correlation results, we have conducted a replication experiment (Exp.1b) and added a test-retest examination to further assess the reliability of our measurements. Specifically, a new group of 24 participants (16 females, 8 males) were recruited to perform the identical experiment procedure as in Experiment 1. Then, after at least seven days, they were asked to return to the lab for a retest. The results successfully replicated the previously reported main effect of emotional condition in both the first test (F(2, 46) = 12.0, p < .001, ηp2 = 0.34, Author response image 1A) and the second test (F(2, 46) = 14.8, p < .001, ηp2 = 0.39, Author response image 1B). The happy BM induced a significantly larger pupil response than the neutral BM (First Test: t(23) = 2.60, p = .022, Cohen’s d = 0.53, 95% CI for the mean difference = [0.02, 0.14], Holm-corrected, p = .048 after Bonferroni correction, Author response image 1A; Second Test: t(23) = 3.36, p = .005, Cohen’s d = 0.68, 95% CI for the mean difference = [0.06, 0.24], Holm-corrected, p = .008 after Bonferroni correction, Author response image 1B). On the contrary, the sad BM induced a significantly smaller pupil response than the neutral BM (First Test: t(23) = -2.77, p = .022, Cohen’s d = 0.57, 95% CI for the mean difference = [-0.19, -0.03], Holm-corrected, p = .033 after Bonferroni correction; Second Test: t(23) = -3.19, p = .005, Cohen’s d = 0.65, 95% CI for the mean difference = [-0.24, -0.05], Holm-corrected, p = .012 after Bonferroni correction, Author response image 1B). Besides, the happy BM induced significantly larger pupil response than the sad BM (first test: t(23) = 4.23, p < .001, Cohen’s d = 0.86, 95% CI for the mean difference = [0.10, 0.28], Holm-corrected, p < .001 after Bonferroni correction, Author response image 1A; second test: t(23) = 4.26, p < .001, Cohen’s d = 0.87, 95% CI for the mean difference = [0.15, 0.44], Holm-corrected, p < .001 after Bonferroni correction, Author response image 1B). The results of the cluster-based permutation analysis were also similar (see Supplementary Material for more details).

**Author response image 1. sa4fig1:** Normalized mean pupil responses in the replication experiment (Experiment 1b) of Experiment 1a and its retest, using the neutral condition as baseline, plotted against happy and sad conditions. (A) In the first test, the group average pupil response to happy intact BM is significantly larger than that to sad and neutral BM, while the pupil response induced by sad BM is significantly smaller than that evoked by neutral BM, replicating the results of Experiment 1a. (B) Moreover, such results were similarly found in the second test.

Notably, we successfully replicated the negative correlation between the happy over sad dilation effect and individual autistic traits in the first test (r(23) = -0.46, p = .023, 95% CI for the mean difference = [-0.73, -0.07], Author response image 2A). No other significant correlations were found (see Author response image 2B-C). Moreover, in the second test, such a correlation was similarly found and was even stronger (r(23) = -0.61, p = .002, 95% CI for the mean difference = [-0.81, -0.27], Author response image 2D). We‘ve also performed a test-retest reliability analysis on the happy over sad pupil dilation effect and the AQ score. The results showed robust correlations. See Author response table 1 for more details.

**Author response table 1. sa4table1:** Reliability of pupil size and AQ indices.

Measurements	First Test	Second Test	Test-retest	
	M(SD)		alpha	r
happy-sad pupil size	0.19(0.22)	0.30(0.34)	0.60	0.61^(****)
AQ score	19.4(5.6)	19.9(6.2)	0.82	0.90^(cdots)

Importantly, in the second test, we’ve also observed a significant negative correlation between AQ and the happy minus neutral pupil dilation effect (r(23) = -0.44, p = .032, 95% CI for the mean difference = [-0.72, -0.04], Author response image 2E), and a significant positive correlation between the sad minus neutral pupil size and AQ (r(23) = 0.50, p = .014, 95% CI for the mean difference = [0.12, 0.75], Author response image 2F). This indicated that the overall correlation between happy over sad dilation effect and AQ was driven both by the diminished happy dilation effect as well as the sad constriction effect. Overall, our replication experiment consistently found a significant negative correlation between AQ and happy over sad dilation effect both in the test and the retest. Moreover, it revealed that such an effect was contributed by both a negative correlation between AQ and happy-neutral pupil response and a positive correlation between AQ and sad-neutral pupil response, demonstrating a general impairment in BM emotion perception (happy or sad) for individuals with high autistic tendencies. This also indicated the utility of adopting a test-retest pupil examination to more precisely detect individual autistic tendencies. We have added these points in the revised text (see lines 135-173, lines 178-180).

**Author response image 2. sa4fig2:** Correlation results for pupil modulation effects and AQ scores in the replication experiment (Experiment 1b) of Experiment 1a and its retest. (A) We replicated the negative correlation between the happy over sad pupil dilation effect and AQ in the first test. (B-C) No other significant correlations were found. (D) In the second test, the negative correlation between the happy over sad pupil dilation effect and AQ was similarly observed and even stronger. (E-F) Moreover, the happy vs. neutral pupil dilation effect and the sad vs. neutral pupil constriction effect respectively correlate with AQ in the second test.

It would be helpful to add discussions as a context to compare the current results with pupil size reactions to emotion signals in picture stimuli.

Thanks for this this thoughtful comment. The modulation of emotional information on pupil responses has been mostly investigated using picture stimuli. Bradley et al. (2008) first demonstrated that humans showed larger pupil responses towards emotional images as compared to neutral images, while no difference was observed between the positive and negative images. This was regarded as the result of increased sympathetic activity induced by emotional arousal that is independent of the emotional valence. Similar results have been replicated with different presentation durations, repetition settings, and tasks (Bradley & Lang, 2015; Snowden et al., 2016). However, the emotional stimuli adopted in these studies were mostly complicated scene images that conveyed rather general emotional information. When it comes to the specific emotion cues (e.g., fear, anger, happy, sad) delivered by our conspecifics through biologically salient signals (e.g., faces, gestures, voices), the results became intermixed. Some studies demonstrated that fearful, disgusted, and angry static faces induced larger pupil sizes than the neutral face, while sad and happy faces failed to induce such pupil dilatory effects (Burley et al., 2017). In contrast, other studies observed larger pupil responses for happy faces as compared to sad and fearful faces (Aktar et al., 2018; Burley & Daughters, 2020; Jessen et al., 2016). These conflicting results could be due to the low-level confounds of emotional faces (e.g., eye size) (Carsten et al., 2019; Harrison et al., 2006). Similar to faces, BM also conveyed salient clues concerning the emotional states of our interactive partners. However, they were highly simplified, deprived of various irrelevant visual confounders (e.g., body shape). Here, we reported that the happy BM induced a stronger pupil response than the neutral and sad BM, lending support to the happy dilation effect observed with faces (Burley & Daughters, 2020; Prunty et al., 2021). Moreover, it helps ameliorate the concern regarding the low-level confounding factors by identifying similar pupil modulations in another type of social signal with distinctive perceptual features. We have added these points to the revised text (see lines 301-321).

References:

Aktar, E., Mandell, D. J., de Vente, W., Majdandžić, M., Oort, F. J., van Renswoude, D. R., Raijmakers, M. E. J., & Bögels, S. M. (2018). Parental negative emotions are related to behavioral and pupillary correlates of infants’ attention to facial expressions of emotion. Infant Behavior and Development, 53, 101–111. https://doi.org/10.1016/j.infbeh.2018.07.004

Bradley, M. M., & Lang, P. J. (2015). Memory, emotion, and pupil diameter: repetition of natural scenes. Psychophysiology, 52(9), 1186–1193. https://doi.org/10.1111/psyp.12442

Bradley, M. M., Miccoli, L., Escrig, M. A., & Lang, P. J. (2008). The pupil as a measure of emotional arousal and autonomic activation. Psychophysiology, 45(4), 602–607. https://doi.org/10.1111/j.1469-8986.2008.00654.x

Burley, D. T., & Daughters, K. (2020). The effect of oxytocin on pupil response to naturalistic dynamic facial expressions. Hormones and Behavior, 125, 104837. https://doi.org/10.1016/j.yhbeh.2020.104837

Burley, D. T., Gray, N. S., & Snowden, R. J. (2017). As far as the eye can see: relationship between psychopathic traits and pupil response to affective stimuli. PLOS ONE, 12(1), e0167436. https://doi.org/10.1371/journal.pone.0167436

Carsten, T., Desmet, C., Krebs, R. M., & Brass, M. (2019). Pupillary contagion is independent of the emotional expression of the face. Emotion, 19(8), 1343–1352. https://doi.org/10.1037/emo0000503

Harrison, N. A., Singer, T., Rotshtein, P., Dolan, R. J., & Critchley, H. D. (2006). Pupillary contagion: central mechanisms engaged in sadness processing. Social Cognitive and Affective Neuroscience, 1(1), 5–17. https://doi.org/10.1093/scan/nsl006

Jessen, S., Altvater-Mackensen, N., & Grossmann, T. (2016). Pupillary responses reveal infants’ discrimination of facial emotions independent of conscious perception. Cognition, 150, 163–169. https://doi.org/10.1016/j.cognition.2016.02.010

Prunty, J. E., Keemink, J. R., & Kelly, D. J. (2021). Infants show pupil dilatory responses to happy and angry facial expressions. Developmental Science, 25(2). https://doi.org/10.11

11/desc.13182

Snowden, R. J., O’Farrell, K. R., Burley, D., Erichsen, J. T., Newton, N. V., & Gray, N. S. (2016). The pupil’s response to affective pictures: role of image duration, habituation, and viewing mode. Psychophysiology, 53(8), 1217–1223. https://doi.org/10.1111/psyp.12668

Overall, I think this is a well-written paper with solid experimental results that support the claim of the authors, i.e., the human visual system may process emotional information in biological motion at multiple levels. Given the key role of emotion processing in normal social cognition, the results will be of interest not only to basic scientists who study visual perception, but also to clinical researchers who work with patients of social cognitive disorders. In addition, this paper suggests that examining pupil size responses could be a very useful methodological tool to study brain mechanisms underlying emotion processing.
**Reviewer #3 (Public Review):**
Summary:The overarching goal of the authors was to understand whether emotional information conveyed through point-light biological motion can trigger automatic physiological responses, as reflected in pupil size.Strengths:This manuscript has several noticeable strengths: it addresses an intriguing research question that fills that gap in existing literature, presents a clear and accurate presentation of the current literature, and conducts a series of experiments and control experiments with adequate sample size. Yet, it also entails several noticeable limitations - especially in the study design and statistical analyses.Weaknesses:(1) Study design:(1.1) Dependent variable:Emotional attention is known to modulate both microsaccades and pupil size. Given the existing pupillometry data that the authors have collected, it would be both possible and valuable to determine whether the rate of microsaccades is also influenced by emotional biological motion.

We thank the reviewer for this advice. Microsaccades functioned as a mechanism to maintain visibility by continuously shifting the retinal image to overcome visual adaptation (Martinez-Conde et al., 2006). Moreover, it was found to be sensitive to attention processes (Baumeler et al., 2020; Engbert & Kliegl, 2003b; Meyberg et al., 2017), and could reflect the activity of superior colliculus (SC) and other related brain areas (Martinez-Conde et al., 2009, 2013). Previous studies have found that, compared with neutral and pleasant images, unpleasant images significantly inhibit early microsaccade rates (Kashihara, 2020; Kashihara et al., 2013). This is regarded as the result of retaining previous crucial information at the sacrifice of updating new visual input. We agree with the reviewer that it would be valuable to investigate whether emotional information conveyed by BM could modulate microsaccades. However, it should be noted that our data collection and experimental design are not optimized for this purpose. This is because we have only recorded the left eye’s data, while abundant methodological studies have doubted the reliability of using only one eye’s data to analyze microsaccades (Fang et al., 2018; Hauperich et al., 2020; Nyström et al., 2017) and suggested that the microsaccades should be defined by spontaneous binocular eye movement (Engbert & Kliegl, 2003a, 2003b). Besides, according to Kashihara et al. (2013), participants showed differential microsaccade rates after the stimuli disappeared so as to maintain the previously observed different emotional information. However, in the current study, we discarded the data after the stimuli disappeared, making it impossible to analyze the microsaccade data after the stimuli disappeared. Despite these disadvantages, we have attempted to analyze the microsaccade rate during the stimuli presentation using only the left eye’s data. Specifically, we applied the algorithm developed by Otero-Millan et al. (2014) (minimum duration = 6 ms, maximum amplitude = 1.5 degrees, maximum velocity = 150 degrees/sec) to the left eye’s data from 100 ms before to 4000 ms after stimulus onset. Subsequently, we calculated the microsaccade rates using a moving window of 100 ms (stepped in 1 ms) (Engbert & Kliegl, 2003b; Kashihara et al., 2013). The microsaccade rate displayed a typical curve, with suppression shortly after stimulus appearance (inhibition phase), followed by an increased rate of microsaccade occurrence (rebound phase). The cluster-based permutation analysis was then applied to explore the modulation of BM emotions on microsaccade rates. However, no significant differences among different emotional conditions (happy, sad, neutral) were found for the four experiments.

**Author response image 3. sa4fig3:** Time-series change in the microsaccade rates to happy, sad, and neutral BM in Experiments 1-4. Solid lines represent microsaccade rates under each emotional condition as a function of time (happy: red; sad: blue; neutral: gray); shaded areas represent the SEM between participants. No significant differences were found after cluster-based permutation correction for the four experiments.

It is important to note that the microsaccade rate analysis was conducted on only the left eye’s data and that the experiment design is not optimized for this analysis, thus, extra caution should be exercised in interpreting the results. Still, we found it very innovative and important to combine the microsaccade index with the pupil size to holistically investigate the processing of emotional information in BM, and future studies are highly needed to adopt more suitable recording techniques and experiment designs to further probe this issue. We have discussed this issue in the revised text (see lines 339-344).

References:

Baumeler, D., Schönhammer, J. G., & Born, S. (2020). Microsaccade dynamics in the attentional repulsion effect. Vision Research, 170, 46–52. https://doi.org/10.1016/j.visres.2020.03.009

Engbert, R., & Kliegl, R. (2003a). Binocular coordination in microsaccades. In The Mind’s Eye (pp. 103–117). Elsevier. https://doi.org/10.1016/b978-044451020-4/50007-4

Engbert, R., & Kliegl, R. (2003b). Microsaccades uncover the orientation of covert attention. Vision Research, 43(9), 1035–1045. https://doi.org/10.1016/s0042-6989(03)00084-1

Fang, Y., Gill, C., Poletti, M., & Rucci, M. (2018). Monocular microsaccades: do they really occur? Journal of Vision, 18(3), 18. https://doi.org/10.1167/18.3.18

Hauperich, A.-K., Young, L. K., & Smithson, H. E. (2020). What makes a microsaccade? a review of 70 years research prompts a new detection method. Journal of Eye Movement Research, 12(6). https://doi.org/10.16910/jemr.12.6.13

Kashihara, K. (2020). Microsaccadic modulation evoked by emotional events. Journal of Physiological Anthropology, 39(1). https://doi.org/10.1186/s40101-020-00238-6

Kashihara, K., Okanoya, K., & Kawai, N. (2013). Emotional attention modulates microsaccadic rate and direction. Psychological Research, 78(2), 166–179. https://doi.org/10.1007/s00426-013-0490-z

Martinez-Conde, S., Macknik, S. L., Troncoso, X. G., & Dyar, T. A. (2006). Microsaccades counteract visual fading during fixation. Neuron, 49(2), 297–305. https://doi.org/10.1016/j.neuron.2005.11.033

Martinez-Conde, S., Macknik, S. L., Troncoso, X. G., & Hubel, D. H. (2009). Microsaccades: a neurophysiological analysis. Trends in Neurosciences, 32(9), 463–475. https://doi.org/10.1016/j.tins.2009.05.006

Martinez-Conde, S., Otero-Millan, J., & Macknik, S. L. (2013). The impact of microsaccades on vision: towards a unified theory of saccadic function. Nature Reviews Neuroscience, 14(2), 83–96. https://doi.org/10.1038/nrn3405

Meyberg, S., Sinn, P., Engbert, R., & Sommer, W. (2017). Revising the link between microsaccades and the spatial cueing of voluntary attention. Vision Research, 133, 47–60. https://doi.org/10.1016/j.visres.2017.01.001

Nyström, M., Andersson, R., Niehorster, D. C., & Hooge, I. (2017). Searching for monocular microsaccades – a red hering of modern eye trackers? Vision Research, 140, 44–54. https://doi.org/10.1016/j.visres.2017.07.012

Otero-Millan, J., Castro, J. L. A., Macknik, S. L., & Martinez-Conde, S. (2014). Unsupervised clustering method to detect microsaccades. Journal of Vision, 14(2), 18–18. https://doi.org/10.1167/14.2.18

(1.2) Stimuli:It appears that the speed of the emotional biological motion stimuli mimics the natural pace of the emotional walker. What is the average velocity of the biological motion stimuli for each condition?

Thanks for pointing out this issue. The neutral and emotional (sad or happy) BM stimuli are equal in walking speed (one step for one second, 1Hz). We have also computed their physical velocity by calculating the Euclidean distance in pixel space of each key point between adjacent frames (Poyo Solanas et al., 2020). The velocity was 5.76 pixels/frame for the happy BM, 4.14 pixels/frame for the neutral BM, and 3.21 pixels/frame for the sad BM. This difference in velocity profile was considered an important signature for conveying emotional information, as the happy walker was characterized by a larger step pace and longer arm swing and the sad walker would instead exhibit a slouching gait with short slow strides and smaller arm movement (Barliya et al., 2012; Chouchourelou et al., 2006; Halovic & Kroos, 2018; Roether et al., 2009). More importantly, our current results could not be explained by the differences in velocities. This is because the inverted emotional BM with identical velocity characteristics failed to induce any modulations on pupil responses. Furthermore, the local sad and happy BM differed the most in velocity feature, while they induced similar modulations on pupil sizes. We have added these points in the revised text (see lines 254-257, 484-491).

References:

Barliya, A., Omlor, L., Giese, M. A., Berthoz, A., & Flash, T. (2012). Expression of emotion in the kinematics of locomotion. Experimental Brain Research, 225(2), 159–176. https://doi.org/10.1007/s00221-012-3357-4

Chouchourelou, A., Matsuka, T., Harber, K., & Shiffrar, M. (2006). The visual analysis of emotional actions. Social Neuroscience, 1(1), 63–74. https://doi.org/10.1080/17470910600630599

Halovic, S., & Kroos, C. (2018). Not all is noticed: kinematic cues of emotion-specific gait. Human Movement Science, 57, 478–488. https://doi.org/10.1016/j.humov.2017.11.008

Poyo Solanas, M., Vaessen, M. J., & de Gelder, B. (2020). The role of computational and subjective features in emotional body expressions. Scientific Reports, 10(1). https://doi.org/10.1038/s41598-020-63125-1

Roether, C. L., Omlor, L., Christensen, A., & Giese, M. A. (2009). Critical features for the perception of emotion from gait. Journal of Vision, 9(6), 15–15. https://doi.org/10.1167/9.6.15

When the authors used inverted biological motion stimuli, they didn't observe any modulation in pupil size. Could there be a difference in microsaccades when comparing inverted emotional biological motion stimuli?

Thanks for this consideration. Both microsaccades and pupil size can provide valuable insights into the underlying neural dynamics of attention and cognitive control (Baumeler et al., 2020; Engbert & Kliegl, 2003; Meyberg et al., 2017). Notably, previous studies have shown that the microsaccades and pupil sizes could be similar and highly correlated in reflecting various cognitive processes, such as multisensory integration, inhibitory control, and cognitive load (Krejtz et al., 2018; Wang et al., 2017; Wang & Munoz, 2021). Moreover, the generation of both microsaccades and pupil responses would involve shared neural circuits, including the midbrain structure superior colliculus (SC) and the noradrenergic system (Hafed et al., 2009; Hafed & Krauzlis, 2012; Wang et al., 2012). However, the pupil size could be more sensitive than microsaccade rates in contexts such as affective priming (Krejtz et al., 2020) and decision formation (Strauch et al., 2018). Moreover, abundant former studies have all shown that inversion would significantly disrupt the perception of emotions from BM (Atkinson et al., 2007; Dittrich et al., 1996; Spencer et al., 2016; Yuan et al., 2022, 2023). Overall, it is unlikely for the microsaccade rates to show significant differences when comparing inverted emotional biological motion stimuli. Besides, we have attempted to analyze the microsaccade rate in the inverted BM situation, while our results showed no significant differences (see also Point 1.1, Author response image 3). Still, it is needed for future studies to combine the microsaccade index and pupil size to provide a thorough understanding of BM emotion processing. We have discussed this issue in the revised text (see lines 339-344).

References:

Atkinson, A. P., Tunstall, M. L., & Dittrich, W. H. (2007). Evidence for distinct contributions of form and motion information to the recognition of emotions from body gestures. Cognition, 104(1), 59–72. https://doi.org/10.1016/j.cognition.2006.05.005

Baumeler, D., Schönhammer, J. G., & Born, S. (2020). Microsaccade dynamics in the attentional repulsion effect. Vision Research, 170, 46–52. https://doi.org/10.1016/j.visres.2020.03.009

Dittrich, W., Troscianko, T., Lea, S., & Morgan, D. (1996). Perception of emotion from dynamic point-light displays represented in dance. Perception, 25(6), 727–738. https://doi.org/10.1068/p250727

Engbert, R., & Kliegl, R. (2003). Microsaccades uncover the orientation of covert attention. Vision Research, 43(9), 1035–1045. https://doi.org/10.1016/s0042-6989(03)00084-1

Hafed, Z. M., Goffart, L., & Krauzlis, R. J. (2009). A neural mechanism for microsaccade generation in the primate superior colliculus. Science, 323(5916), 940–943. https://doi.org/10.1126/science.1166112

Hafed, Z. M., & Krauzlis, R. J. (2012). Similarity of superior colliculus involvement in microsaccade and saccade generation. Journal of neurophysiology, 107(7), 1904-1916.

Krejtz, K., Duchowski, A. T., Niedzielska, A., Biele, C., & Krejtz, I. (2018). Eye tracking cognitive load using pupil diameter and microsaccades with fixed gaze. Plos One, 13(9), e0203629. https://doi.org/10.1371/journal.pone.0203629

Krejtz, K., Żurawska, J., Duchowski, A., & Wichary, S. (2020). Pupillary and microsaccadic responses to cognitive effort and emotional arousal during complex decision making. Journal of Eye Movement Research, 13(5). https://doi.org/10.16910/jemr.13.5.2

Meyberg, S., Sinn, P., Engbert, R., & Sommer, W. (2017). Revising the link between microsaccades and the spatial cueing of voluntary attention. Vision Research, 133, 47–60. https://doi.org/10.1016/j.visres.2017.01.001

Spencer, J. M. Y., Sekuler, A. B., Bennett, P. J., Giese, M. A., & Pilz, K. S. (2016). Effects of aging on identifying emotions conveyed by point-light walkers. Psychology and Aging, 31(1), 126–138. https://doi.org/10.1037/a0040009

Strauch, C., Greiter, L., & Huckauf, A. (2018). Pupil dilation but not microsaccade rate robustly reveals decision formation. Scientific Reports, 8(1). https://doi.org/10.1038/s41598-018-31551-x

Wang, C.-A., Blohm, G., Huang, J., Boehnke, S. E., & Munoz, D. P. (2017). Multisensory integration in orienting behavior: pupil size, microsaccades, and saccades. Biological Psychology, 129, 36–44. https://doi.org/10.1016/j.biopsycho.2017.07.024

Wang, C.-A., Boehnke, S. E., White, B. J., & Munoz, D. P. (2012). Microstimulation of the monkey superior colliculus induces pupil dilation without evoking saccades. Journal of Neuroscience, 32(11), 3629–3636. https://doi.org/10.1523/jneurosci.5512-11.2012

Wang, C.-A., & Munoz, D. P. (2021). Differentiating global luminance, arousal and cognitive signals on pupil size and microsaccades. European Journal of Neuroscience, 54(10), 7560–7574. https://doi.org/10.1111/ejn.15508

Yuan, T., Ji, H., Wang, L., & Jiang, Y. (2022). Happy is stronger than sad: emotional information modulates social attention. Emotion. https://doi.org/10.1037/emo0001145

Yuan, T., Wang, L., & Jiang, Y. (2023). Cross-channel adaptation reveals shared emotion representation from face and biological motion. In Emotion (p. In Press).

(2) Statistical analyses(2.1) Multiple comparisons:There are many posthoc comparisons throughout the manuscript. The authors should consider correction for multiple comparisons. Take Experiment 1 for example, it is important to note that the happy over neutral BM effect and the sad over neutral BM effect are no longer significant after Bonferroni correction, which is worth noting.

Thanks for this suggestion. In our original analysis, we applied the Holm post-hoc corrections for multiple comparisons. The Holm correction is a step-down correction method and is more powerful but less conservative than the Bonferroni correction. We have now conducted the stricter Bonferroni post-hoc correction. In Experiment 1, the happy over neutral, and happy over sad BM effect is still significant after the Bonferroni post-hoc correction (happy vs. neutral: p = .036; happy vs. sad: p = .009), and the sad over neutral comparison remains marginally significant after the Bonferroni post-hoc correction (p = .071). Importantly, the test-retest replication experiment also yielded significant results for the comparisons between happy and neutral (First Test: p = .022, Holm-corrected, p = .048, Bonferroni-corrected; Second Test: p = .005, Holm-corrected, p = .008, Bonferroni-corrected), sad and neutral (First Test: p = .022, Holm-corrected, p = .033, Bonferroni-corrected; Second Test: p = .005, Holm-corrected, p = .012, Bonferroni-corrected, Author response image 1B), and happy and sad BM (First test: p < .001, Holm-corrected, p < .001, Bonferroni-corrected; Second test: p < .001, Holm-corrected, p < .001, Bonferroni-corrected). These results provided support for the replicability and consistency of the reported significant contrasts. See also Point 2.3.

In Experiment 4, the significance levels of all comparisons remained the same after Bonferroni post-hoc correction (happy vs. neutral: p = .011; sad vs. neutral: p = .007; happy vs. sad: p = 1.000). We have now added these results in the main text (See lines 119, 122, 124, 143, 145, 148, 150, 153, 155, 248, 251, 254).

(2.2) The authors present the correlation between happy over sad dilation effect and the autistic traits in Experiment 1, but do not report such correlations in Experiments 2-4. Did the authors collect the Autistic Quotient measure in Experiments 2-4? It would be informative if the authors could demonstrate the reproducibility (or lack thereof) of this happy-sad index in Experiments 2-4.

We apologize for not making it clear. We have collected the AQ scores in Experiments 2-4. However, it should be pointed out that the happy over sad pupil dilation effect was only observed in Experiment 1. Moreover, we’ve again identified such happy over sad pupil dilation effect in the replication experiment (Experiment 1b) as well as its correlation with AQ. Instead, no significant correlations between AQ and the happy-sad pupil index were found in Experiments 2-4, see Author response image 4 for more details. We have reported these correlations in the main text (see lines 157-173, 190-194, 212-216, 257-262).

**Author response image 4. sa4fig4:** Correlations between the happy over sad pupil dilation effect and AQ scores. (A) The happy over sad pupil dilation effect correlated negatively with individual autistic scores. (B-C) Such correlation was similarly observed in the test and retest of the replication experiment. (D-F) No such correlations were found for the inverted, nonbiological, and local BM stimuli.

(2.3) The observed correlation between happy over sad dilation effect and the autistic traits in Experiment 1 seems rather weak. It could be attributed to the poor reliability of the Autistic Quotient measure or the author-constructed happy-sad index. Did the authors examine the test-retest reliability of their tasks or the Autistic Quotient measure?

Thanks for this suggestion. We have now conducted a test-retest replication study to further confirm the observed significant correlations. Specifically, we recruited a new group of 24 participants (16 females, 8 males) to perform the identical procedure as in Experiment 1, and they were asked to return to the lab for a retest after at least seven days. We’ve replicated the significant main effect of emotional conditions in both the first test (F(2, 46) = 12.0, p < .001, ηp2 = 0.34) and the second test (F(2, 46) = 14.8, p < .001, ηp2 = 0.39). Besides, we also replicated the happy minus neutral pupil dilation effect (First Test: t(23) = 2.60, p = .022, Cohen’s d = 0.53, 95% CI for the mean difference = [0.02, 0.14], Holm-corrected, p = .048 after Bonferroni correction; Second Test: t(23) = 3.36, p = .005, Cohen’s d = 0.68, 95% CI for the mean difference = [0.06, 0.24], Holm-corrected, p = .008 after Bonferroni correction), and the sad minus neutral pupil constriction effect (First Test: t(23) = -2.77, p = .022, Cohen’s d = 0.57, 95% CI for the mean difference = [-0.19, -0.03], Holm-corrected, p = .033 after Bonferroni correction; Second Test: t(23) = -3.19, p = .005, Cohen’s d = 0.65, 95% CI for the mean difference = [-0.24, -0.05], Holm-corrected, p = .012 after Bonferroni correction). Additionally, the happy BM still induced a significantly larger pupil response than the sad BM (first test: t(23) = 4.23, p < .001, Cohen’s d = 0.86, 95% CI for the mean difference = [0.10, 0.28], Holm-corrected, p < .001 after Bonferroni correction; second test: t(23) = 4.26, p < .001, Cohen’s d = 0.87, 95% CI for the mean difference = [0.15, 0.44], Holm-corrected, p < .001 after Bonferroni correction).

Notably, we’ve successfully replicated the negative correlation between the happy over sad dilation effect and individual autistic traits (r(23) = -0.46, p = .023, 95% CI for the mean difference = [-0.73, -0.07]). Such a correlation was similarly found and was even stronger in the retest (r(23) = -0.61, p = .002, 95% CI for the mean difference = [-0.81, -0.27]). A test-retest reliability analysis was conducted on the happy over sad pupil dilation effect and the AQ score. The results showed robust correlations (r(happy-sad pupil size) = 0.56; r(AQ) = 0.90) and strong test-retest reliabilities (α(happy-sad pupil size) = 0.60; α(AQ) = 0.82). We have added these results to the main text (see lines 135-173). See also Response to Reviewer #2 Response 1 for more details.

(2.4) Relatedly, the happy over sad dilation effect is essentially a subtraction index. Without separately presenting the pipul size correlation with happy and sad BM in supplemental figures, it becomes challenging to understand what's primarily driving the observed correlation.

Thanks for pointing this out. We have now presented the separate correlations between AQ and the pupil response towards happy and sad BM in Experiment 1 (see Author response image 5A), and the test-retest replication experiment of Experiment 1 (see Author response image 5B-C). No significant correlations were found. This is potentially because the raw pupil response is a mixed result of BM perception and emotion perception, while the variations in pupil sizes across emotional conditions could more faithfully reflect individual sensitivities to emotions in BM (Burley et al., 2017; Pomè et al., 2020; Turi et al., 2018).

**Author response image 5. sa4fig5:** No significant correlations between AQ and pupil response towards happy and sad intact BM were found in Experiment 1a and the test-retest replication experiment (Experiment 1b).

To probe what's primarily driving the observed correlation between happy-sad pupil size and AQ, we instead used the neutral as the baseline and separately correlated AQ with the happy-neutral and the sad-neutral pupil modulation effects. No significant correlation was found in Experiment 1a (Author response image 6A-B) and the first test of the replication experiment (Experiment 1b) (Author response image 6C-D). Importantly, in the second test of the replication experiment, we found a significant negative correlation between AQ and the happy-neutral pupil size (r(23) = -0.44, p = .032, 95% CI for the mean difference = [-0.72, -0.04], Author response image 6E), and a significant positive correlation between AQ and the sad-neutral pupil size (r(23) = 0.50, p = .014, 95% CI for the mean difference = [0.12, 0.75], Author response image 6F). This suggested that the overall correlation between AQ and the happy over sad dilation effect was driven by diminished pupil modulations towards both the happy and sad BM for high AQ individuals, demonstrating a general deficiency in BM emotion perception (happy or sad) among individuals with high autistic tendencies. It further revealed the potential of adopting a test-retest pupil examination to more precisely detect individual autistic tendencies. We have reported these results in the main text (see lines 166-173).

**Author response image 6. sa4fig6:** Correlation results for pupil modulations and AQ scores. (A-B) In Experiment 1a, no significant correlation was observed between AQ and the happy pupil modulation effect, as well as between AQ and the sad pupil modulation effect. (C-D) Similarly, no significant correlations were found in the first test of the replication experiment (Experiment 1b). (E-F) Importantly, in the second test of Experiment 1b, the happy vs. neutral pupil dilation effect was positively correlated with AQ, and the sad vs. neutral pupil constriction effect was positively correlated with AQ.

References:

Burley, D. T., Gray, N. S., & Snowden, R. J. (2017). As Far as the Eye Can See: Relationship between Psychopathic Traits and Pupil Response to Affective Stimuli. PLOS ONE, 12(1), e0167436. https://doi.org/10.1371/journal.pone.0167436

Pomè, A., Binda, P., Cicchini, G. M., & Burr, D. C. (2020). Pupillometry correlates of visual priming, and their dependency on autistic traits. Journal of vision, 20(3), 3-3.

Turi, M., Burr, D. C., & Binda, P. (2018). Pupillometry reveals perceptual differences that are tightly linked to autistic traits in typical adults. eLife, 7. https://doi.org/10.7554/elife.32399

(2.5) For the sake of transparency, it is important to report all findings, not just the positive results, throughout the paper.

Thanks for this suggestion. We have now reported all the correlations results between AQ and pupil modulation effects (happy-sad, happy-neutral, sad-neutral) in the main text (see lines 130-131, 157-162, 166-170, 190-194, 212-216, 257-262). Given that no significant correlations were observed between AQ and the raw pupil responses across four experiments, we reported their correlations with AQ in the supplementary material. We have stated this point in the main text (see lines 132-134).

(3) Structure(3.1) The Results section immediately proceeds to the one-way repeated measures ANOVA. This section could be more reader-friendly by including a brief overview of the task procedures and variables, e.g., shifting Fig. 3 to this section.

Thanks for this advice. We have now added a brief overview of the task procedures and variables and we have also shifted the figure position (see lines 101-103).

**Reviewer #1 (Recommendations For The Authors):**
(1) I suggest that the authors first explain the task (i.e., Fig. 3) at the beginning of the results. And it seems more appropriate to show the time course figures (Fig. 2) and before the bar plots (Fig. 1). If I understand correctly, the bar plots reflect the averaged data from the time course plots. Also, please clearly state the time window used to average the data. The results of the correlation analysis can be displayed in the last step.

Thanks for this suggestion. We have now added a concise explanation of the task at the beginning of the results (see lines 101-103). We have also adjusted the figure positions and adjusted the order of our results according to the reviewer’s suggestion. The time window we used to average the data was from the onset of the stimuli until the end of the stimuli presentation. We have now clearly stated these issues in the revised text (see lines 111-112).

(2) According to the above, I think a more reasonable arrangement should be Fig. 3, 2, and 1.

Thanks for this suggestion. We have adjusted the figure positions accordingly.

(3) Please include each subject's data points in the bar plots in Fig. 1.

We have now presented each subject’s individual data point in the bar plot.

(4) Lines 158-160 and 199-202 report interaction effects of the two-way ANOVA. This is good, but the direction of interaction effect should also be reported.

We thank the reviewer for this suggestion. We have now reported the direction of the interaction effect. The significant interaction observed across Experiment 1 and Experiment 2 was mainly due to the diminishment of emotional modulation in inverted BM. The significant interaction crossing Experiment 1 and Experiment 3 was similarly caused by the lack of emotional modulation in nonbiological stimuli. With regard to the significant interaction across Experiment 1 and Experiment 4, it could be primarily attributed to the vanishment of pupil modulation effect between happy and sad local BM. We have specified these points in the revised text, see lines 198-199, 219-220, 267-269.

**Reviewer #3 (Recommendations For The Authors):**
(1) Number of experiments:As stated in the Methods section, this study seems to consist of five experiments (120/24=5) according to the description below. However, the current manuscript only reports findings from four of these experiments. Can the authors clarify on this matter?"A total of 120 participants (44 males, 76 females) ranging from 18 to 29 years old (M ± SD = 23.1 ± 2.5) were recruited, with 24 in each experiment."

We apologize for not making it clear. This referred to a pure behavior explicit emotion classification experiment (N=24) that served as a prior test to confirm that the local BM stimuli conveyed recognizable emotional information. We have now more carefully stated this issue in the revised text, see lines 456-458.

(2) Emotion processing mechanism of BM"Mechanism" is a very strong word, suggesting a causal relationship. In the setting of a passive viewing task that lacks any behavioral report, it is possible that the observed changes in pupil size could be epiphenomenal, rather than serving as the underlying mechanism.

Thanks for this suggestion. We have now either changed “mechanism” into “phenomenon” or deleted it. We have also carefully discussed the potential implications for future studies to incorporate variant behavioral, physiological and neural indexes to yield more robust causal evidence to unveil the potential mechanism serving the observed multi-level BM emotion processing phenomenon.

(3) Data sharingThe authors could improve their efforts in promoting data transparency to ensure a comprehensive view of the results. This implies sharing deidentified raw data instead of summary data in an Excel spreadsheet.

Thanks for this suggestion. We have now uploaded the deidentified raw data. (https://doi.org/10.57760/sciencedb.psych.00125).